# Microtubule-binding protein MAP1B regulates interstitial axon branching of cortical neurons via the tubulin tyrosination cycle

Jakub Ziak[1], Joelle M Dorskind[1,2], Brian Trigg[1], Sriram Sudarsanam[1], Xinyu O Jin [ID][1], Randal A Hand[1,3] & Alex L Kolodkin [ID][1✉]

## Abstract

Regulation of directed axon guidance and branching during development is essential for the generation of neuronal networks. However, the molecular mechanisms that underlie interstitial (or collateral) axon branching in the mammalian brain remain unresolved. Here, we investigate interstitial axon branching in vivo using an approach for precise labeling of layer 2/3 callosal projection neurons (CPNs). This method allows for quantitative analysis of axonal morphology at high acuity and also manipulation of gene expression in well-defined temporal windows. We find that the GSK3β serine/threonine kinase promotes interstitial axon branching in layer 2/3 CPNs by releasing MAP1B-mediated inhibition of axon branching. Further, we find that the tubulin tyrosination cycle is a key downstream component of GSK3β/MAP1B signaling. These data suggest a cell-autonomous molecular regulation of cortical neuron axon morphology, in which GSK3β can release a MAP1B-mediated brake on interstitial axon branching upstream of the posttranslational tubulin code.

**Keywords** Interstitial Axon Branching; Cortical Neuron Development; Microtubules; Intracellular Signaling
**Subject Categories** Cell Adhesion, Polarity & Cytoskeleton; Development; Neuroscience

## Introduction

The cerebral cortex is a six-layered brain structure that is essential for sensory processing, motor skills, learning and memory, and thought. The establishment of functional cortical connectivity requires newly born neurons to migrate into appropriate layers during embryogenesis and then elaborate complex and unique axon and dendrite branching patterns. This results in laminar-specific organization of interstitial axon branches and also the elaboration of select intra- and inter-areal cortical connections. Cortical excitatory projection neurons possess long axons and connect to target neurons in other cortical and subcortical regions (Winnubst et al, 2019). Since a single neuron often projects to multiple targets, both extensive and precise axon branching patterns are required to provide distinct inputs to specific brain regions (Chédotal and Richards, 2010; Greig et al, 2013). Given the complexity of axon trajectories, precise regulation of collateral axon branch formation is vital for generation of a functional brain connectome (Armijo-Weingart and Gallo, 2017). However, the intracellular signaling pathways, receptors, cell surface molecules and extracellular ligands that regulate laminar-specific interstitial axon branching in the cortex are poorly understood.

Two major types of axonal branching are growth cone splitting and interstitial (collateral) branching (Gibson and Ma, 2011; Kornack and Giger, 2005). Though growth cone splitting is typical for peripheral neurons, a significant fraction of cortical neuron axon branches are generated by interstitial branching: a new axon collateral growing from a stable, previously established, axon shaft (Armijo-Weingart and Gallo, 2017; Kalil and Dent, 2014). It can take several days until a daughter branch is formed from a primary axon. For example, axons of excitatory layer 2/3 callosal projection neurons (CPNs) develop from postnatal day 0 (P0) in mice. However, their first collateral branches begin to be established starting at ~P3 in the ipsilateral cortex, a time when the growth cone of the primary axon has already crossed the CNS midline (Hand et al, 2015). Therefore, cytoskeletal components within the axon shaft must be locally rearranged at the site of the axonal branch point, followed by the stabilization of newly formed microtubules (MTs) that support the growth of the nascent branch (Kalil and Dent, 2014). Much previous work in vitro has identified components of axon branching in cortical neurons and also dorsal root ganglion (DRG) neurons, and a major challenge is to relate these observations to in vivo axon branching (Dorskind and Kolodkin, 2021; Hoersting and Schmucker, 2021; Dorskind et al, 2023). The factors that contribute to regulation of axon collateral branching in vitro include different MT-associated proteins (MAPs) (MAP7, MAP1B), MT severing proteins spastin and katanin, molecules that regulate actin (drebrin, Arp2) and various enzymes (including GSK3β). For example, in cultured DRG neurons, the MT-associated protein 7 (MAP7) prevents nascent axon branches

[1]The Solomon H. Snyder Department of Neuroscience, Johns Hopkins Kavli Neuroscience Discovery Institute, The Johns Hopkins School of Medicine, 725 North Wolfe St., Baltimore, MD 21205, USA. [2]Present address: Novartis Institutes for BioMedical Research, Boston, MA, USA. [3]Present address: Prilenia Therapeutics, Boston, MA, USA.
✉E-mail: kolodkin@jhmi.edu

from retracting, while the MT-associated protein 1B (MAP1B) restricts axon branch initiation (Tymanskyj et al, 2017; Barnat et al, 2016). In addition, mitochondria and the endoplasmic reticulum localize to the base of developing axonal branches and contribute to their stabilization (Courchet et al, 2013; Spillane et al, 2013). However, the molecular mechanisms that regulate interstitial axon branching in vivo remain to be elucidated.

The underlying molecular basis of central nervous system (CNS) interstitial axon branching has been studied in vivo in *Drosophila*, *Xenopus*, and Zebrafish models, (Cantallops et al, 2000; Kim et al, 2007; Drinjakovic et al, 2010; Urwyler et al, 2019; Izadifar et al, 2021; Martin et al, 2024) owing in part to available genetic labeling strategies that allow for precise and dynamic visualization of individual neurons. However, translation of these observations to mammals is challenging since the experimental approaches that allow for studying mouse neuronal connectivity in a quantitative fashion at the single cell level are limited (Fenlon and Richards, 2015). Currently, some of the most promising tools include certain viral-based labeling strategies which, however, are not ideal for developmental studies due to the long period of time (days to weeks) necessary for viral-mediated gene expression (Economo et al, 2019; Ueda et al, 2020). Further, the immense diversity of neuronal subtypes in the mammalian CNS, including subtypes with unique morphologies and distinct neural process trajectories, makes the identification of the molecular pathways that direct unique axon branching patterns in individual neuronal subtypes in vivo challenging.

We have recently developed tools (Dorskind et al, 2023) that overcome many of these difficulties, allowing for temporally controlled targeting of small populations of layer 2/3 CPNs and quantitative assessment of axon branching patterns. Importantly, most layer 2/3 CPNs have a highly conserved bimodal interstitial branching pattern; their axons form no, or very few, primary interstitial axon branches within layer 4 and only branch robustly in layer 5, where they form ~6–8 primary interstitial branches (Hand et al, 2015). The few layer 2/3 CPN interstitial axon branches that do form in layer 4 rarely extend within this layer and instead grow apically or basally to innervate layer 2/3 or layer 5, respectively. Taken together, layer 2/3 CPNs in the mouse cortex provide a useful model system to uncover molecular mechanisms that regulate interstitial axon branching in the mammalian CNS.

Here, we show that activation of the serine threonine kinase GSK3β induces excessive branching in layer 2/3 CPNs, and we identify one of its targets, MAP1B, as a cell-autonomous branch restricting factor. In addition, we find that GSK3β/MAP1B signaling regulates the tyrosination/detyrosination cycle of α-tubulin which in turn increases or decreases the probability of generating interstitial axon branches in layer 2/3 CPNs. Together, these data describe a cell-autonomous molecular pathway that regulates interstitial axon branching in mammalian cortical neurons.

# Results

## Activation of GSK3β in excitatory cortical neurons promotes interstitial axon branching

To investigate the molecular mechanisms that govern interstitial axon branching in neocortical neurons, we recently performed an in utero electroporation-based candidate screen in layer 2/3 CPNs (Dorskind et al, 2023) (Fig. EV1A). We chose signaling molecules, axon guidance receptors and cytoskeletal regulators known to play roles in axon branching elsewhere in the nervous system, designing dominant negative (DN) constructs, overexpression (OE) DNA constructs, shRNAs and guide RNAs (gRNAs) to perturb their function in vivo using IUE. One of the strongest phenotypes we observed resulted from overexpression of human constitutively active GSK3β kinase, which harbors the S9A mutation (GSK3β-CA (Eldar-Finkelman et al, 1996)) (Dorskind et al, 2023). Axonal morphologies of these neurons were significantly altered (Fig. EV1B,C); we observed increased interstitial branching in layer 4, extensive looping and twisting of aberrant axon branches in the cortex, altered axon trajectories within the corpus callosum, and generation of shorter protrusions along layer 2/3 axons. This initial observation was surprising since previous in vitro studies using DRG neurons, and also in vivo studies on peripheral nerves, showed an opposite effect of GSK3β activity on axon branching (Tokuoka et al, 2002; Zhou et al, 2004; Kim et al, 2006; Zhao et al, 2009; Bilimoria et al, 2010). These studies suggested that inhibition of GSK3β activity increases the probability of axonal branch formation, however, they were not specifically focused on interstitial axon branching. Given that growth cone splitting is a major mode of peripheral nerve axon branching, (Gibson and Ma, 2011) our current results suggest that GSK3β regulates axon branching in a context-dependent manner. Furthermore, morphological analysis of cultured hippocampal neurons isolated from *Gsk3β*$^{-/-}$ mice did not show changes in neurite numbers (Kim et al, 2006).

To investigate GSK3β in the context of neocortical collateral axon branching, we prepared a new murine GSK3β-CA mutant construct and overexpressed it in layer 2/3 CPNs using our recently developed tamoxifen-dependent bipartite system that allows for sparse (~1–100 neurons/brain) and robust neuronal labeling along with the capacity to perform genetic manipulations via in utero electroporation (Dorskind et al, 2023) (Fig. 1A). Our method utilizes a combination of two recombinases, tamoxifen-dependent Cre (encoded by a *pCAG-Cre-ERT2* plasmid) and Cre-dependent FlpO (*pEF1-Flex-FlpO*), that sequentially activates expression of a fluorophore from a third plasmid (*pCAG-FSF-mCherry* or *pCAG-FSF-GFP*). This approach permits examination of individual axon morphologies and quantitative morphological assessments at single cell resolution. Indeed, overexpression of mouse GSK3β-CA (using a GFP-expressing and FlpO-dependent *pCAG-FSF-GSK3β-CA-IRES-GFP* plasmid; see Methods and Table EV1 for experimental details) at embryonic day 15.5 (E15.5) in layer 2/3 CPNs resulted in a phenotype comparable to our original observations with human GSK3β (Dorskind et al, 2023) (Fig. 1B). We quantified layer 2/3 neuron primary axon collaterals in layers 4 and 5 at postnatal day 14 (P14) and found their numbers were significantly increased in comparison to control brains, displayed here in cumulative frequency plots (Fig. 1B). These plots represent relative cumulative distribution (percentages) of neurons according to the extent of their interstitial axon branching; neurons with more branches shift these curves to the right. Data points represent number of interstitial branches determined for all neurons analyzed across all individual animals. For clarity, all data are also presented in the form of scatter plots depicting absolute number of axon branches in individual neurons from individual animals, together with the mean number of branches and standard deviations (Appendix

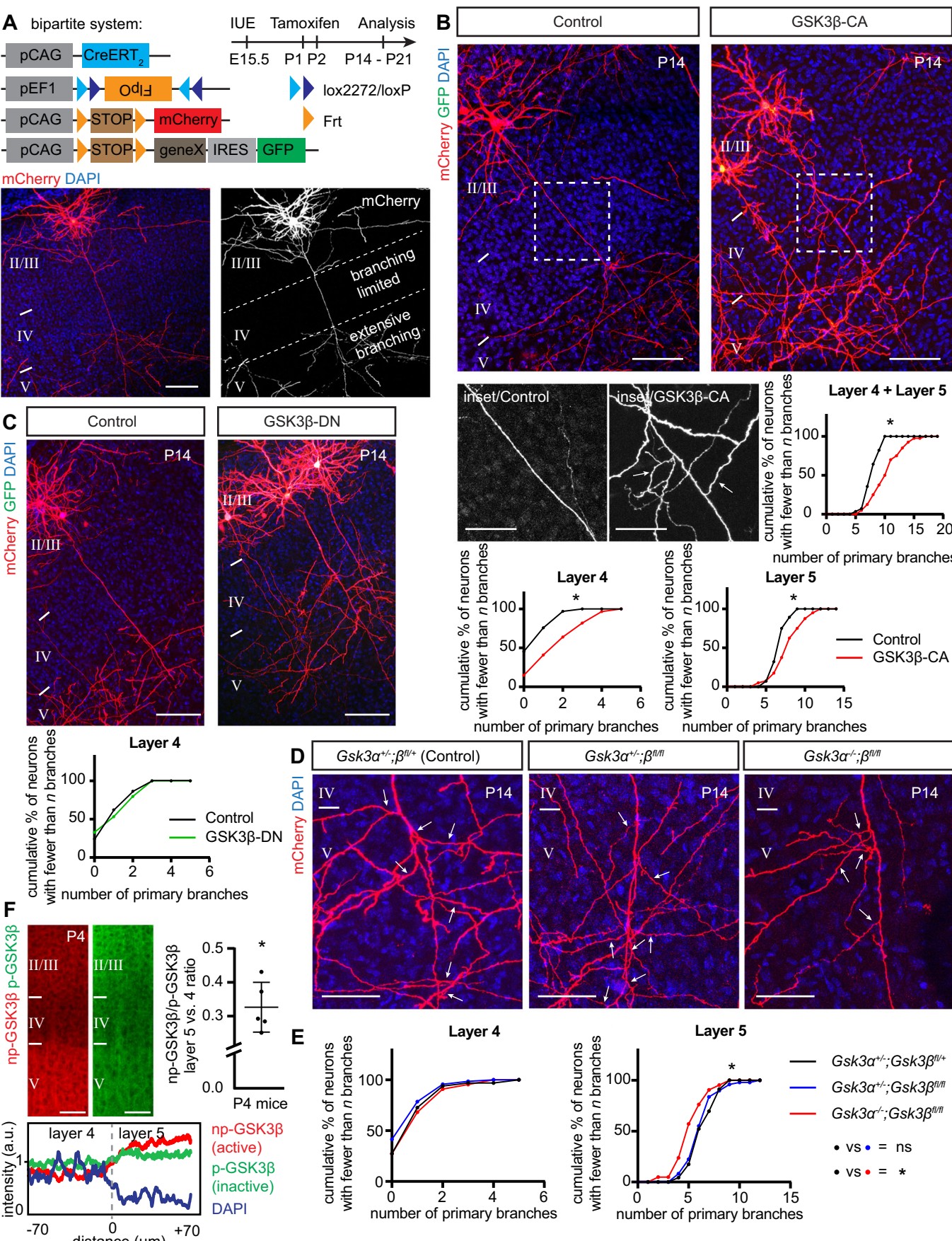

◄

**Figure 1. Activation of GSK3β in layer 2/3 callosal projection neurons promotes interstitial axon branching.**

(A) Bipartite system for sparse gene expression and neuronal labeling in the layer 2/3 CPNs. Plasmids (*pCAG-CreERT₂, pEF1-Flex-FlpO, pCAG-Frt-STOP-Frt-mCherry*, and optionally *pCAG-Frt-STOP-Frt-geneX-IRES-GFP*) were in utero electroporated at E15.5 and tamoxifen was administered into the milk spot of the P1 and P2 pups. Brains were analyzed at P14 or P21 (note time stamp in all images). An example of an individual sparsely labeled layer 2/3 neuron is shown in lower panels. The DAPI signal was used to identify individual cortical layers (II/III, IV and V) in all experiments. See Table EV1 for detailed information about plasmids used in all experiments. Scale bars: 100 μm. (B) Overexpression of constitutively active GSK3β induces interstitial axon branching in layer 2/3 CPNs. Confocal images of representative neurons are shown, with insets from axonal segments passing through layer 4 (arrows, ectopic interstitial branches). Primary interstitial axon branches were quantified in layers 4 and 5. Data are presented as a relative cumulative distribution of neurons with the indicated number of axonal branches. A curve shifted to the right represents increased branching and vice versa. The same graphical representation is used throughout all following figures. See Appendix for detailed branching quantification for every sample in the form of scatter dot plots. For the description of statistics, see Materials/Methods. Data were fitted into the Mixed Effects Model (Maxwell et al, 2018) and analyzed with either a *t*-test or a one-way ANOVA. Table EV1 contains absolute *p* values for all comparisons. *$p < 0.05$, t-test. Scale bars: 100 μm, 50 μm (inset). (C) Overexpression of dominant-negative GSK3β does not lead to changes in interstitial axon branching. Branches were quantified in layer 4, and data analysis is the same as in panel (B). Scale bars: 100 μm. (D, E) GSK3α and GSK3β are necessary for the generation of interstitial axon branches. Cell-autonomous deletion of *Gsk3β* (using *pCAG-CreERT₂, pEF1-Flex-FlpO, pCAG-Frt-STOP-Frt-mCherry* plasmids) does not influence axonal branching, however removal of both *Gsk3α* and *Gsk3β* leads to a decrease in the number of primary interstitial axon branches in cortical layer 5. The insets show axonal segments passing through layer 5 (arrows, interstitial branches). Data analysis and presentation is the same as in panel (A). *$p < 0.05$, ANOVA with post hoc Dunnet's test. Scale bars: 50 μm. (F) Activation of GSK3β in the cortex. Cortical slices from P4 mice were probed with non-phospho-GSK3β (np-GSK3β, active enzyme, red) and phosoho-GSK3β (pS9-GSK3β, inactive enzyme, green) antibodies. A sample trace of intensity measurement is shown at the bottom. Right: np-GSK3β/p-GSK3β intensity in cortical layer 4 and layer 5 was determined using a Paired t test and is presented as a layer 5/layer 4 ratio (individual values from $n = 5$ mice are shown, mean ± S.D. is plotted). *$p < 0.05$. Scale bar: 50 μm. Source data are available online for this figure.

Figs. S1–S5, S10; also, see Fig. 1 Legend and Materials and Methods for the fitting of data to the Mixed Effects Model and the statistical tests that were applied to these data to determine significance). In addition, we analyzed the effect of GSK3β-CA expression in layer 2/3 CPNs in 3-months old mice and found results similar to those we observed at P14 (Fig. EV1D), indicating that these ectopic branches become a stable component of cortical axon projections.

Next, we asked if GSK3β enzymatic activity is required for the regulation of collateral axon branching by overexpressing catalytically inactive (dominant negative) GSK3β, which harbors the K85R mutation (GSK3β-DN (Sun et al, 2008)). We did not observe increased branching in layer 4 following GSK3β-DN overexpression (Fig. 1C), indicating that GSK3β kinase activity is sufficient to induce layer 2/3 CPN ectopic collateral axon formation in layer 4. To test if GSK3β is necessary for interstitial axon branching, we utilized conditional *Gsk3β^{fl/fl}* mice (Morgan-Smith et al, 2014) to delete GSK3β specifically in layer 2/3 CPNs using IUE and our bipartite labeling and expression system (Fig. 1D). We performed these *Gsk3β^{fl/fl}* IUE-mediated single neuron knock-out experiments in *Gsk3α^{−/−}* and *Gsk3α^{+/−}* genetic backgrounds to account for GSK3α compensatory effects. We observed no changes in collateral axon branching across cortical layers in *Gsk3β* knockout layer 2/3 neurons but found significant decreases in the number of primary layer 5 collateral axon branches in *Gsk3α/β* double knockouts (Figs. 1D,E and EV1E). We noted that in *GSK3α^{−/−}* mutants there were no changes in axon branching across cortical layers (Fig. EV1F). These results show that both GSK3α and GSK3β kinases are positive regulators of layer 2/3 CPN interstitial axon branching. In agreement with previous findings, (Morgan-Smith et al, 2014) we also observed that the patterning of basal dendrites is disrupted in *Gsk3α/β* double knockouts such that basal dendrites extend towards pial surface rather than laterally (Fig. EV1G, H). Finally, we assessed the distribution of activated (non-phosphorylated, np-GSK3β) and inactivated (phosphorylated, p-GSK3β) kinase in neocortical layers at P4. We found that the ratio of np-GSK3β/p-GSK3β is higher in layer 5, where interstitial branching occurs, than in layer 4, where it does not (Fig. 1F).

Taken together, these results in cortical excitatory neurons in vivo reveal an unexpected and previously unrecognized role for GSK3β kinase in the regulation of interstitial axon branching. Activation of GSK3β is sufficient to promote interstitial axon branching by layer 2/3 CPNs, and both GSK3α and GSK3β are necessary for the development of collateral axon branches.

## MAP1B restricts layer 2/3 CPN axon interstitial branching

What are the downstream targets that GSK3β regulates to mediate interstitial axon branching in layer 2/3 excitatory neurons? GSK3β serves a hub for more than 100 molecules, (Sutherland, 2011) including regulators of gene expression, protein synthesis, cell cycle regulation, neuronal differentiation and regulation of cell morphology through modification of the cytoskeleton. Since we find that GSK3β-CA overexpression leads to neuronal process morphological phenotypes, we focused on known GSK3β phosphorylation targets with the capacity to influence neuronal cytoskeletal dynamics. These include members of both canonical and noncanonical Wnt signaling pathways, such as MAP1B, MACF1, CLASP1, CLASP2, APC and β-catenin, all of which can be directly phosphorylated by GSK3β (Sutherland, 2011). We utilized either conditional knockout mice (*Apc^{fl/fl}, β-catenin^{fl/fl}*) (Bruxvoort et al, 2007; Brault et al, 2001) or previously characterized shRNAs (Benoist et al, 2013; Ka et al, 2014; Mimori-Kiyosue et al, 2005) that we cloned into the *pPrime-dsRed-miR30* vector (Fig. EV2A) to enable direct control of expression in individual targeted layer 2/3 neurons. From this survey we found that knockdown of MAP1B in layer 2/3 CPNs causes ectopic collateral axon branching in layer 4 (Fig. 2A,B), suggesting that MAP1B acts to restrict axon branching. MAP1B is a cytosolic protein that binds to multiple cytoskeletal components (Villarroel-Campos and Gonzalez-Billault, 2014). Although the MAP1B actin- and microtubule-binding domains were experimentally verified previously, (Hammarback et al, 1991; Cueille et al, 2007) the MAP1B structure has not been solved nor reliably predicted (Fig. EV2E). We did not observe any changes in interstitial axon branching for any of the other candidate genes we tested (Fig. EV2C,D). In addition to these shRNA-mediated loss-of-function experiments, we validated our MAP1B results using a CRISPR/Cas9 approach (Fig. EV2B). IUE using Cas9 and sgRNAs

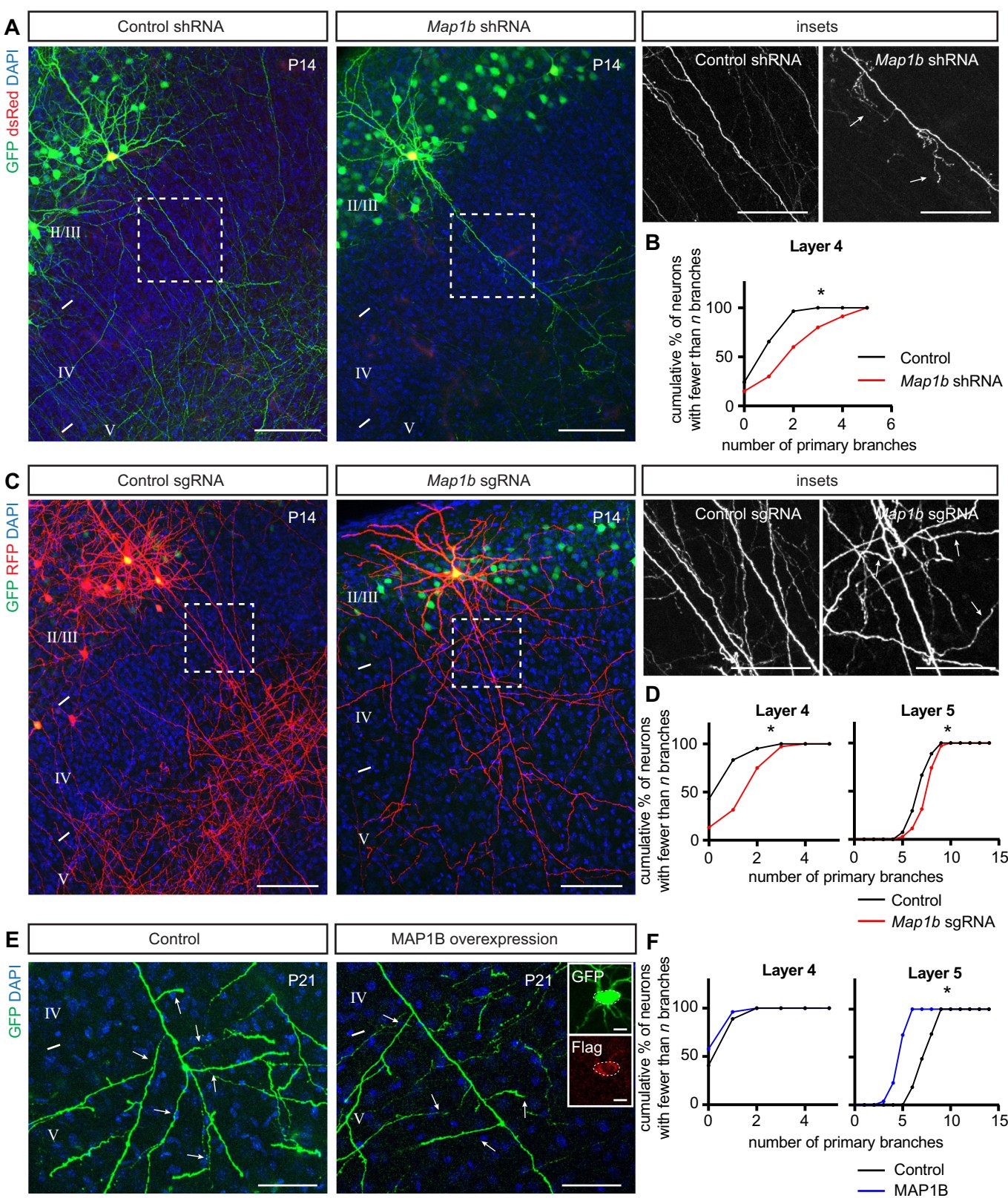

**Figure 2. MAP1B restricts layer 2/3 CPN axon interstitial branching.**

(A, B) Knockdown of *Map1b* using shRNA IUE leads to the generation of aberrant interstitial axon branches in layer 4 (arrows, right insets). Data analysis and presentation as in Fig. 1. *p < 0.05, t-test. Scale bars: 100 µm, 50 µm (insets). (C, D) Knockout of *Map1b* using CRISPR/Cas9 IUE leads to the generation of aberrant interstitial axon branches in cortical layers 4 and 5 (arrows, insets on right). Data analysis and presentation is the same as in Fig. 1. *p < 0.05, *t*-test. Scale bars: 100 µm, 50 µm (insets). (E, F) Gain-of-function (GOF) experiments with a full-length Flag-tagged MAP1B leads to a reduction in interstitial branching in layer 5 (arrows). Inset on the right shows GFP and Flag signals in the cell body of analyzed neurons. Data analysis and presentation as in Fig. 1. *p < 0.05, *t*-test. Scale bars: 50 µm, 10 µm (inset on right). Source data are available online for this figure.

that recognize four different exons within the *Map1b* locus resulted in similar ectopic interstitial axon branching patterns in layer 4 and, in addition, ectopic interstitial axon branching in layer 5 (Fig. 2C,D).

We next investigated the localization of MAP1B in developing layer 2/3 axons using the recently developed targeted knock-in with two guides (TKIT) method for endogenous protein tagging mediated by CRISPR/Cas9 mutagenesis (Fang et al, 2021) (Fig. EV3A). We designed two guide RNAs directed towards the 5' and 3' introns flanking the targeted *Map1b* exon (exon 7) and provided a donor exon sequence fused to GFP with a short linker. TKIT plasmids (*pX330* encoding the two guides and Cas9, and pMini carrying the donor sequence) were electroporated in utero at E15.5 together with an mCherry expression plasmid to label targeted neurons. As a proof of concept, we tagged endogenous actin (exon 2) and observed robust localization of GFP in dendritic spines, well known actin-rich structures (Fig. EV3B). Using this approach, we observed that MAP1B is highly expressed at P4 (around the time when interstitial axon branching begins in layer 2/3 CPNs (Hand et al, 2015)) and localized in the cytoplasm of dendrites and axons, as well as in growing axon collaterals and distal axons in the corpus callosum (Fig. EV3C). Similar results were found after tagging endogenous GSK3β (exon 11, Fig. EV3D).

Since our endogenous tagging showed that MAP1B is not restricted to any axonal compartment, we next asked if its overexpression can inhibit collateral axon branching. We overexpressed full length C-terminal Flag-tagged MAP1B in layer 2/3 CPNs using our bipartite system and quantified collateral axon branching in layer 5. Indeed, electroporated neurons showed a ~30% reduction in the number of primary intestinal axonal branches in layer 5 (Fig. 2E,F) compared to control neurons. There were no changes in axon branching in layer 4, as expected (Fig. 2F). Together, these data show that MAP1B is expressed in developing layer 2/3 cortical axons and restricts interstitial axon branching of these CPNs.

## MAP1B acts downstream of GSK3β to regulate axon collateral branching

MAP1B is phosphorylated by multiple kinases including GSK3β, Dyrk1a and casein kinase 1 (CK1) (Villarroel-Campos and Gonzalez-Billault, 2014; Scales et al, 2009). There are at least three residues within the MAP1B microtubule-association domain (MTA; Fig. EV2E) that are recognized by GSK3β: two unprimed (S1260, T1265) and one (S1391) that is primed by Dyrk1a phosphorylation at S1395 (Scales et al, 2009). MAP1B phosphorylated at all of these sites is known as "mode I phosphorylated MAP1B" and has been implicated in promoting axon outgrowth (Trivedi et al, 2005). Interestingly, previous experiments showed that MAP1B is poorly phosphorylated when overexpressed alone

(e.g. without a GSK3β kinase) in heterologous non-neuronal cells in vitro (Goold et al, 1999). Since our data suggest that overexpression of MAP1B restricts axon branching (Fig. 2E), we hypothesized that this effect is mediated by non-phosphorylated (naïve) MAP1B. We therefore asked whether GSK3β phosphorylation of MAP1B is necessary for collateral axon branching by taking advantage of point mutations in MAP1B phosphoresidues. We prepared a dephosphomimetic (non-phosphorylatable) MAP1B mutant harboring the mutations S1260A, T1265A and S1391A (referred to here as MAP1B-ΔP), and also a phosphomimetic MAP1B mutant harboring S1260D, T1265D and S1391D mutations (referred to here as MAP1B-P). We tested the effects of these MAP1B mutants on layer 2/3 CPN interstitial branching compared to overexpression of naïve MAP1B. We found that neurons expressing MAP1B-ΔP had a similar number of interstitial axon branches in layers 4 and 5 as neurons expressing wild-type MAP1B (Fig. 3A,B). On the other hand, overexpression of the MAP1B-P phosphomimetic gain-of-function mutant protein led to a significant increase in interstitial axon branching in layers 4 and 5, as compared to wild-type MAP1B (Fig. 3A,B), indicating that phosphorylated MAP1B promotes interstitial axon branching. Direct comparison with control WT neurons showed that phosphomimetic MAP1B-P induces ectopic axon branches in layer 4 (Fig. EV3E).

Next, we investigated the distribution of phosphorylated MAP1B (pThr1265-MAP1B) in the P4 cortex and found that it is enriched in deeper cortical layers (Fig. EV3F), consistent with the idea that phosphorylated MAP1B promotes interstitial axon branching. Thus, our results gained from GSK3β and MAP1B manipulations suggest that GSK3β is a MAP1B activating enzyme. Therefore, we hypothesized that preventing MAP1B phosphorylation should rescue the GSK3β-CA phenotype. We overexpressed GSK3β-CA either alone or together with our dephosphomimetic MAP1B-ΔP construct and quantified the number of primary axon branches in layer 4 and layer 5. Indeed, we found that overexpression of MAP1B-ΔP is epistatic to GSK3β-induced ectopic interstitial axon branching (Fig. 3C,D). This suppression was not complete, since we detected the formation of small protrusions along some axons despite the presence of MAP1B-ΔP (Fig. 3C, arrowheads), suggesting that another GSK3β substrate is involved in this process. Supporting this notion, our data show that overexpression of MAP1B-P does not fully phenocopy the GSK3β-CA phenotype and leads to increased branching in layer 4, but not in layer 5, as compared to WT neurons (Fig. EV3E). Alternatively, it is also possible that the maximal level of MAP1B-P that can be obtained using IUE does not meet the threshold necessary for promoting ectopic branching in layer 5. Taken together, these results show that MAP1B restricts interstitial axon branching in its non-phosphorylated state, and that these axonal branch-restrictive properties of MAP1B (a 'MAP1B brake' on

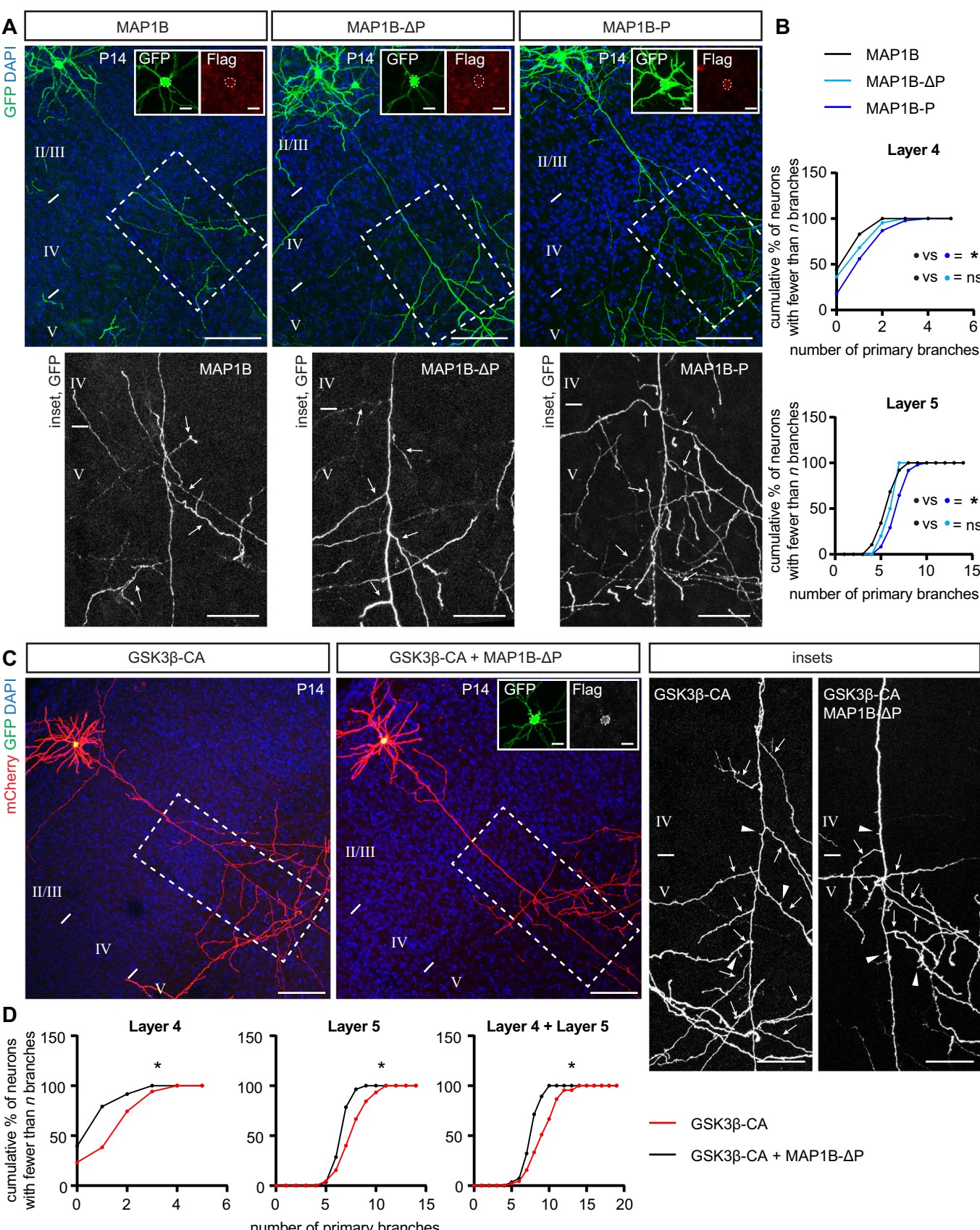

◄ **Figure 3.  Branch-restrictive properties of MAP1B are released following GSK3β phosphorylation.**

(A, B) Overexpression of Flag-tagged MAP1B, dephosphomimetic MAP1B mutant (MAP1B S1260A, T1265A, S1391A, referred to here as MAP1B-ΔP) and phosphomimetic MAP1B mutant (MAP1B S1260D, T1265D, S1391D, referred as MAP1B-P) in layer 2/3 CPNs. Insets show details of neuronal cell bodies with GFP and Flag signals. Insets at bottom show interstitial axon branches forming in layer 5 (arrows). MAP1B-ΔP has a similar effect on branching as wild-type MAP1B, while MAP1B-P increases the number of axonal branches in comparison with wild-type MAP1B. Data analysis and presentation as in Fig. 1. *$p < 0.05$, ANOVA with post hoc Dunnet's test. Scale bars: 100 µm, 50 µm (insets at bottom), 10 µm (cell body insets). (C, D) MAP1B-ΔP rescues the GSK3β-CA phenotype. Overexpression of GSK3β-CA together with dephosphomimetic MAP1B-ΔP reduces the overall number of interstitial axon branches compared to GSK3β-CA alone. Inset shows GFP and Flag signals in the neuronal cell body. Insets on right show segments of layer 4 and 5 with arrows marking interstitial axon branches. Note that short protrusions (axon segments shorter than 30 µm) are formed under both conditions (arrowheads). Data analysis and presentation as in Fig. 1. *$p < 0.05$, $t$-test. Scale bars: 100 µm, 50 µm (insets), 10 µm (cell body insets). Source data are available online for this figure.

interstitial axon branches) are released following GSK3β phosphorylation.

## α-tubulin tyrosination promotes collateral axon branching

Our data show that GSK3β promotes interstitial axon branching through phosphorylation of MAP1B, while naïve dephosphorylated MAP1B restricts it. This prompted us to investigate the link between GSK3β-mediated MAP1B phosphorylation and cytoskeletal changes that could regulate collateral axon branching. Shifting the balance of MAP1B towards its phosphorylated form increases the ratio of tyrosinated:detyrosinated (tyr/detyr) MTs, (Goold et al, 1999) and our results suggest that MAP1B phosphorylation underlies the formation of axon collaterals. Therefore, we hypothesized that promotion of interstitial axon branching requires higher levels of tyrosinated MTs, in contrast to increased levels of detyrosinated MTs that would restrict branching. This further suggests that total tyrosinated MT levels will be higher in layer 2/3 axons as they pass through layer 5, where interstitial axon branching occurs, as compared to layer 4, where it does not. We first tested this idea by immunostaining cortical sections using antibodies that specifically recognize either tyrosinated or detyrosinated tubulins (see Methods). We found that the tyr/detyr ratio is indeed higher in the neuropil of layer 5 than the neuropil of layer 4 in the P4 cortex (Appendix Fig. S6A). Further, the distribution of total α-tubulin, and also tubulin polyglutamylation and K40 acetylation, in the developing cortex is independent of cortical lamination (Fig. EV4A; Appendix Fig. S6B,C) and GSK3β activation (Appendix Fig. S7A,B). All tubulin modifications we investigated are present in axons, and tyrosinated tubulin is also prominent in apical dendrites of cortical pyramidal neurons (Appendix Fig. S6D) (Janke and Magiera, 2020).

To gain a more precise assessment of tyrosinated MT distribution, we used a recently designed fluorescent TagRFP-based sensor (Tyr-T sensor) that preferentially binds to tyrosinated tubulin (Kesarwani et al, 2020) and expressed it in individual layer 2/3 CPNs to directly measure MT tyrosination. We used the CRISPR/Cas9-based TKIT method combined with plasmid delivery by in utero electroporation, as described above, employing CRISPR guides to target the endogenous mouse *Rosa26* locus and also providing a donor plasmid that encodes the Tyr-T sensor sequence (Fig. 4A). We co-electroporated a GFP-expressing construct to identify IUE-targeted regions in the cortex, harvested the brains at P4, and analyzed tyrosinated MT distribution in individual neurons. We immunolabeled slices derived from these electroporated cortices with α-tubulin antibody and quantified the ratio of TagRFP/α-tubulin fluorescence along individual layer 2/3 axons in

layer 4 and layer 5 at P4 (neuronal compartments with high concentrations of tyrosinated MTs will have a higher TagRFP/α-tubulin ratio). A validation experiment using overexpression of enzymes that regulate the tubulin tyrosination cycle confirmed the functionality of this sensor (Fig. EV4B). Only neurons with brighter cell bodies showed detectable sensor fluorescence in axons (Fig. EV4C), so therefore we focused on these for our analyses. The TagRFP$^+$ signal was normalized to the α-tubulin signal from the same axonal segment to calculate the RFP/α-tubulin ratio. Comparing this parameter between layer 4 and layer 5 in individual layer 2/3 neuron axons revealed a significant increase in the RFP/α-tubulin ratio in layer 2/3 CPN axons within layer 5, as compared to axon segments within layer 4 (Fig. 4B); this indicates that the distribution of tyrosinated tubulin in the axons of layer 2/3 CPNs is biased towards layer 5. Thus, these results show a positive correlation between the MT tyrosination signal and the layer-specific position of layer 2/3 CPN axon segments in the cortical wall, counteracting the normally occurring tubulin detyrosination in axonal segments of neurons, which is normally enriched in interstitial axon branches.

We next addressed the functional relationship between tubulin tyrosination and interstitial axon branching. Using our bipartite system, we overexpressed either full length TTL (tyrosine tubulin ligase) or detyrosinating enzymes (VASH1 or VASH2) together with SVBP (small vasohibin binding protein). SVBP acts as a chaperon and cofactor for VASH1 and for VASH2 (Aillaud et al, 2017). These approaches led to expected changes in the level of tyrosinated tubulin: a decrease in MT tyrosination with VASH1/SVBP and an increase in MT tyrosination with TTL, as revealed by immunolabeling of brain slices derived from experimental cortices (Fig. EV5A). Quantification of interstitial axon branching showed that overexpression of TTL increases interstitial axon branching in layers 4 and 5, while VASH1/SVBP had no effect (Fig. 4C–E). This effect of TTL is likely mediated by its enzymatic activity since overexpression of dominant-negative TTL mutant (TTL-DN, harboring K198A and D200A mutations (Prota et al, 2013; Szyk et al, 2011) in layer 2/3 CPNs significantly decreased branching in layer 4 as compared to naïve TTL overexpression (Fig. 4F,G). These results suggest that increasing the level of tubulin tyrosination is sufficient to promote interstitial axon branching in cortical neurons.

## Disrupting the tubulin tyrosination cycle causes aberrant interstitial axon branching

The role of the tubulin tyrosination cycle in interstitial axon branching is unexplored (Sanyal et al, 2023). Previous experiments have focused on analysis of axon outgrowth in immature neuronal

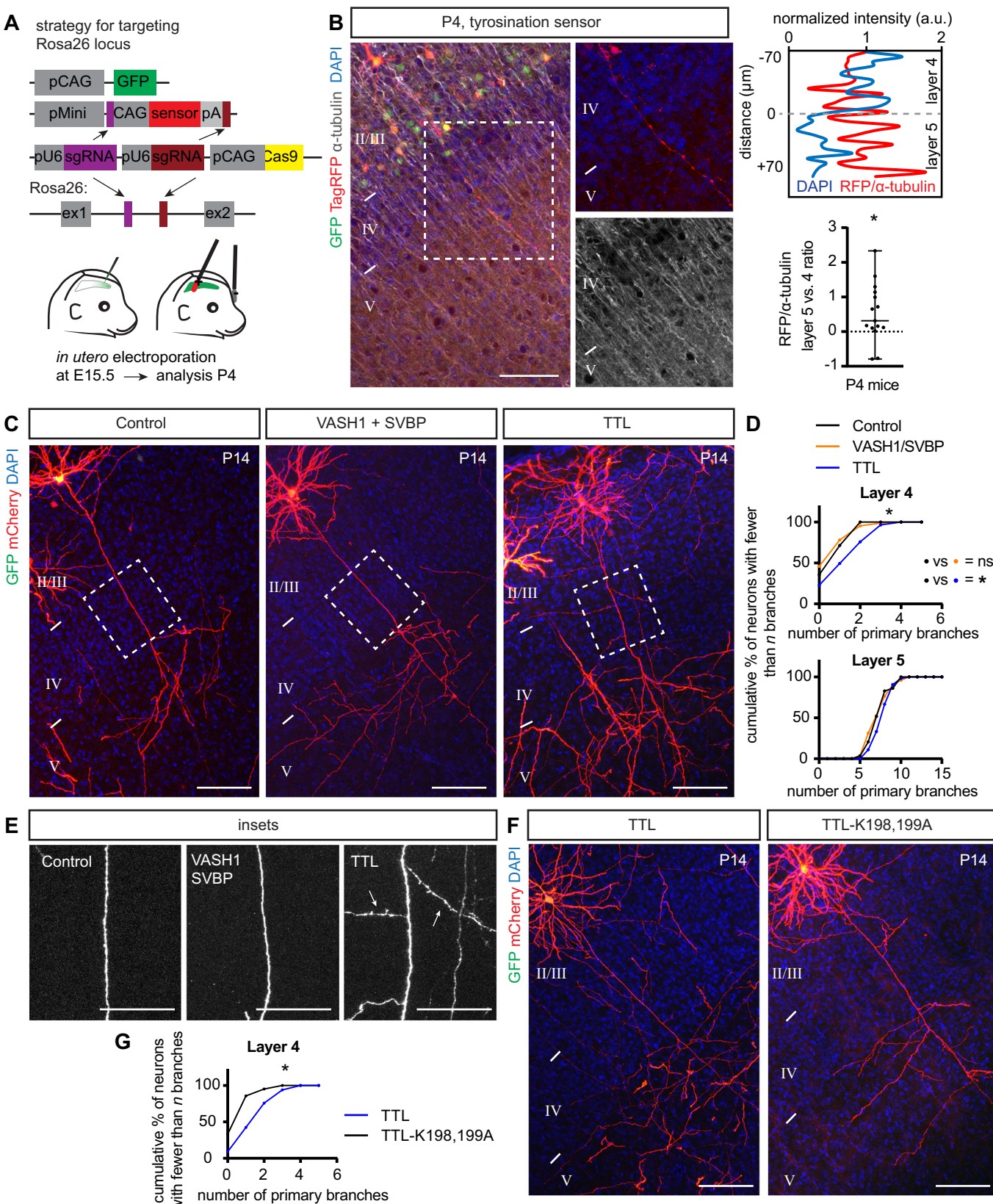

**Figure 4.  α-tubulin tyrosination promotes collateral axon branching.**

(A) Strategy for analysis of tyrosinated α-tubulin distribution using a Tag-RFP sensor that recognizes tyrosinated tubulin and a CRISPR/Cas9 knock-in approach (see main text for sensor details). Plasmids were delivered at E15.5 via *in utero* electroporation, the sensor was expressed from the *Rosa26* locus following targeted insertion, and the tyrosination signal was analyzed at P4. (B) An example of tyrosinated tubulin sensor distribution along layer 2/3 CPN axons. The Tag-RFP signal in layer 4 and layer 5 was normalized to α-tubulin and is presented as a normalized intensity. A sample trace is shown at the top right, where the blue line denotes the DAPI signal and the red line denotes normalized RFP/α-tubulin intensity. Bottom right: RFP/α-tubulin intensity in axons passing through layer 4 and layer 5 was compared using a Wilcoxon Matched-pairs test and is presented as a layer 5/layer 4 ratio (individual values with means and min/max error bars are plotted, $n = 15$ neurons from 4 mice). *$p < 0.05$. Scale bar: 100 µm. (C) Gain of function experiments with VASH1 and SVBP (to promote detyrosination) and TTL (to promote tyrosination) show that TTL overexpression is sufficient to promote interstitial axon branching. Promoting detyrosination did not influence interstitial axon branch formation. Scale bars: 100 µm. (D) Quantification of the effect of VASH1/SVBP and TTL overexpression on interstitial axon branching. Data analysis and presentation as in Fig. 1. *$p < 0.05$, ANOVA with post hoc Dunnet's test. (E) Insets from (C) highlight axon segments within layer 4. Note ectopic branches formed in layer 4 following TTL overexpression (arrows). Scale bars: 50 µm. (F, G) Overexpression of a dominant-negative TTL (TTL-K198,199 A) does not lead to the formation of ectopic interstitial axon branches in layer 2/3 CPNs, indicating that enzymatic activity of TTL is necessary to for this process. Data analysis and presentation as in Fig. 1. *$p < 0.05$, t-test. Scale bars: 100 µm. Source data are available online for this figure.

cultures (Erck et al, 2005; Iqbal et al, 2019). However, contrasting results have been obtained from $Ttl^{-/-}$ neurons and $Svbp^{-/-}$ neurons. Embryonic hippocampal or precerebellar neurons isolated from embryonic $Ttl^{-/-}$ mice show an increased axon growth rate and neurite branching after 2 days in vitro (DIV2) (Erck et al, 2005; Marcos et al, 2009). Similarly, embryonic hippocampal neurons isolated from $Svbp^{-/-}$ mice display more neurite branches in vitro at DIV2 (Aillaud et al, 2017). $Ttl^{-/-}$ mice die within a day after birth, and so analysis of interstitial axon branching in vivo in these mice is not possible (Erck et al, 2005). Neuronal morphology in $Svbp^{-/-}$ or the recently generated $Matcap^{-/-}$ mice, which both lack detyrosinated microtubules, has not yet been investigated (Land-skron et al, 2022; Pagnamenta et al, 2019). Our results from gain-of-function experiments targeting the tubulin tyrosination cycle suggest that increasing tubulin tyrosination promotes interstitial axon branching.

To address the necessity of tubulin tyrosination for interstitial axon branching, we used a CRISPR mutagenesis strategy to generate mutations that result in the deletion of enzymes that modulate the tubulin tyrosination cycle in vivo in individual layer 2/3 neurons. We designed CRISPR guides targeting the first four exons of *Ttl* or the first three exons of *Svbp*, respectively. Control staining with an antibody against tyrosinated MTs confirmed the efficiency of this strategy at the level of single neurons (Fig. EV5B). Deletion of *Ttl*, which shifts the tyrosination/detyrosination ratio toward detyrosination, did not lead to increased or decreased axon branching, although we noted a marked increase in dendritic branching in targeted neurons (Fig. 5A). However, deletion of *Svbp*, which favors tyrosination, resulted in an increase in layer 2/3 CPN interstitial axon branching in layer 4, but not in layer 5 (Fig. 5A,B). We also noted a mild defect in neuronal migration of mutated cells (Appendix Fig. S7C), which is consistent with previous observations (Aillaud et al, 2017). Further, analysis of $Svbp^{-/-}$ mice showed that the depletion of detyrosinated tubulin is incomplete in these mutants, (Pagnamenta et al, 2019) suggesting the existence of additional enzyme(s) with detyrosinating activity. Indeed, a recent study identified the MATCAP protein, which also removes tyrosine residues from tubulin (Landskron et al, 2022). Therefore, we simultaneously deleted both *Svbp* and *Matcap*, however this manipulation did not lead to a further increase in interstitial axon branching since we observed a phenotype comparable to that observed following *Svbp* deletion alone (Fig. EV5C–E). Never-theless, our loss-of-function experiments suggest that disruption of the tyrosination/detyrosination ratio causes aberrant interstitial axon branching in layer 2/3 CPNs.

## GSK3β/MAP1B signaling regulates interstitial axon branching through modification of microtubule dynamics and tubulin tyrosination

Previous in vitro studies showed that tyrosinated microtubules are more dynamic than detyrosinated microtubules (Kreis, 1987; Schulze and Kirschner, 1987). Since our data indicate that promoting tubulin tyrosination supports interstitial axon branch-ing, we assessed microtubule dynamics in developing layer 2/3 CPN axons. E15.5 embryos were in utero electroporated with pCAG-mCherry and pCAG-EB3-GFP. At E18.5, neurons were dissociated from electroporated cortices and plated on glass-bottom cell culture plates. This approach decreases phenotypic variability of trans-fected neurons since the majority of electroporated neurons represent layer 2/3 CPNs. At DIV4, EB3-GFP was tracked under acute GSK3β pharmacological inhibition (using the CHIR99021 inhibitor, (Bennett et al, 2002) Fig. 6A). We found that GSK3β inhibition decreases the speed of EB3-GFP at developing branch points (Fig. 6A,B, Appendix Fig. S8, Movie EV1). These results show that inhibition of GSK3β decreases microtubule dynamics, which is in line with our proposed model.

Since we find that GSK3β signaling regulates interstitial axon branching by releasing an inhibitory MAP1B brake on axon branching that results in the promotion of tubulin tyrosination, we next asked if there is indeed a link between GSK3β activity and tubulin tyrosination. If alteration of the tubulin tyrosination/detyrosination cycle occurs downstream of a MAP1B brake on the regulation of axon branching, activation of GSK3β should promote tyrosination; similarly, blocking tyrosination via TTL deletion should prevent formation of GSK3β-induced interstitial axon branches. To test these ideas, we over-expressed GSK3β-CA together with CRISPR/Cas9 plasmids targeting TTL. As a control, GSK3β-CA was co-electroporated with control CRISPR/Cas9 constructs. We found that removing TTL suppresses aberrant GSK3β-CA-induced interstitial axon branch formation in layer 4 and layer 5 (Fig. 6C,D). Next, we investigated the interaction between the MAP1B brake on interstitial axon branching and tyrosination using CRISPR-mediated loss-of-function. We deleted either *Map1b* alone (control), or together with *Ttl* to inhibit tyrosination (Fig. 6E). These manipulations showed that preventing tubulin tyrosination normalizes ectopic interstitial axon branching induced by MAP1B deficiency (Fig. 6F). Taken together, these results show that TTL acts downstream from the GSK3β/MAP1B signaling axis, demonstrating how regulated posttranslational MT modifications (the tubulin code) contribute to stereotypic patterns of CPN interstitial axon branching.

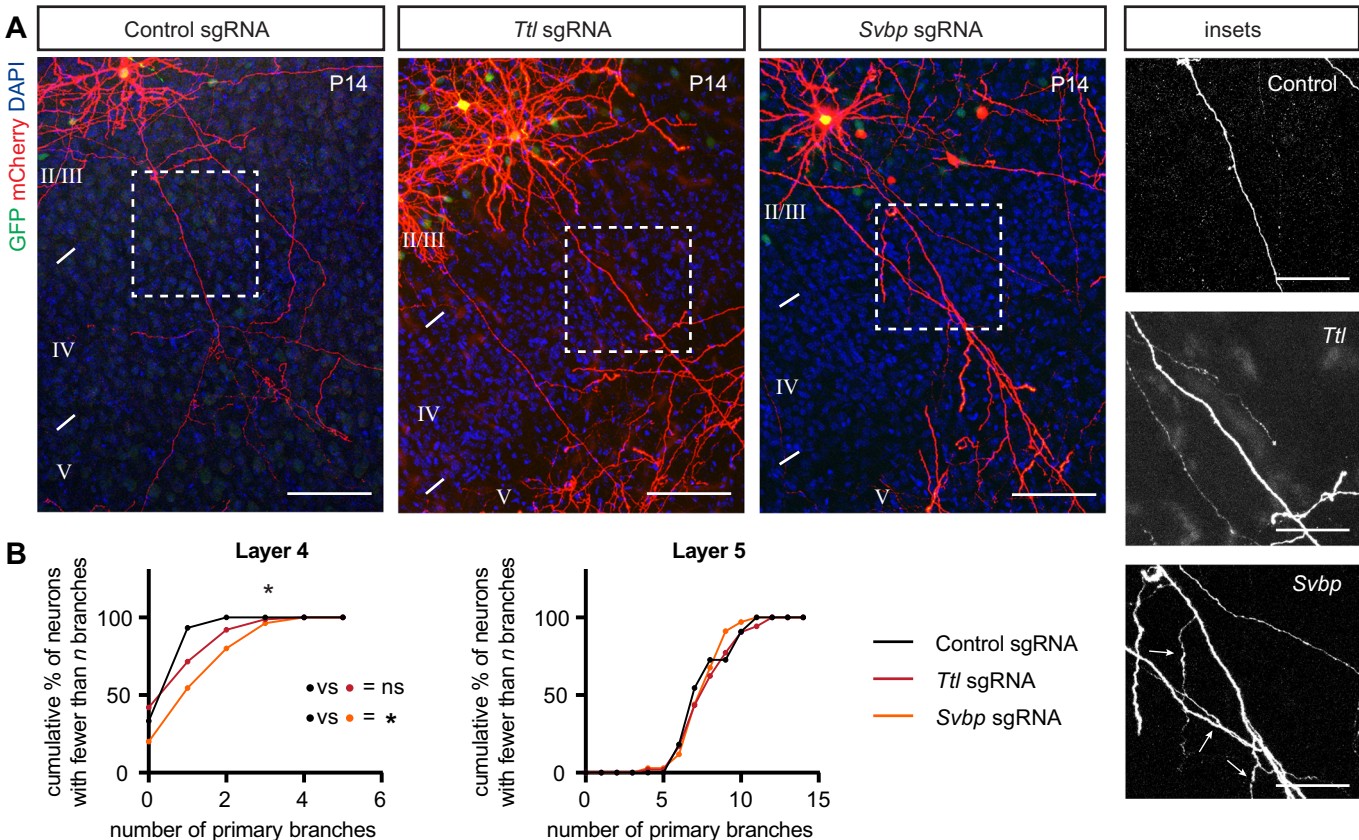

**Figure 5. Disrupting the tubulin tyrosination cycle causes aberrant interstitial axon branching.**

(A, B) Loss-of-function of enzymes regulating tyrosination/detyrosination cycle using a CRISPR/Cas9 approach. *Svbp* deletion increases the number of primary axon branches in layer 4. *Ttl* deletion does not affect interstitial axon branching. Insets on right show ectopic axon branches in axons passing through layer 4 following removal of *Svbp* (arrows). Data analysis and presentation as in Fig. 1. *$p < 0.05$, ANOVA with post hoc Dunnet's test. Scale bars: 100 μm, 50 μm (insets). Source data are available online for this figure.

# Discussion

Interstitial axon branching is a fundamental phenomenon, allowing CNS neurons to connect to multiple targets that are spatially distinct. In the neocortex, laminar-specific interstitial axon branching provides a scaffold for the elaboration of select cortical circuitry. Defining molecular mechanisms that regulate interstitial axon branching in the CNS remains a major unresolved question. Here, we use recently developed robust methods (Dorskind et al, 2023) to study axonal morphology in vivo in mammalian cortical neurons, and we quantitatively analyze axonal morphology of thousands of neurons at a single-cell resolution to reveal cytosolic regulators of interstitial axon branching in mammalian layer 2/3 CPNs. A major finding from our experiments is that the MT binding protein MAP1B acts to restrict interstitial CPN axon branching in vivo, and this brake on branching can be released following GSK3β-mediated phosphorylation. Subsequently, the ratio of tyrosinated/detyrosinated tubulin shifts towards the tyrosinated form and promotes formation of interstitial axon branches (see Fig. 7 for a graphical summary). These conclusions are supported by the following: (1) GSK3β loss-of-function and gain-of-function experiments show that GSK3α and β isoforms are necessary, and GSK3β is sufficient, for interstitial axon branching;

(2) MAP1B loss-of-function, overexpression and mutagenesis experiments show that naïve unphosphorylated MAP1B restricts interstitial branching while phosphorylated MAP1B promotes it; (3) Genetic manipulations of the tubulin tyrosination cycle reveal that increasing the ratio of tyrosinated to detyrosinated α-tubulin promotes interstitial axon branching; and (4) Decreasing tyrosination using CRISPR-mediated loss-function prevents excessive formation of interstitial axon branches induced by GSK3β gain-of-function or MAP1B loss-of-function. Together, these observations reveal an intracellular signaling pathway that cell-autonomously regulates interstitial axon branching in mammalian cortical neurons.

The identification of signaling pathways that regulate interstitial axon branching so as to regulate cortical laminar-specific innervation has been elusive, though there are multiple signaling molecules known to direct cortical neuronal axon pathfinding and axon extension in cortical neuron subtypes. These include classical axon guidance cues, morphogens, and their receptors (reviewed in: Armijo-Weingart and Gallo, 2017; Kalil and Dent, 2014; Dorskind and Kolodkin, 2021; Hoersting and Schmucker, 2021; Vanderhaeghen and Polleux, 2023; Bradke, 2022). In addition, cytosolic signaling molecules, including Ankyrin-B, (Creighton et al, 2021) are capable of refining the formation of terminal contralateral CPN

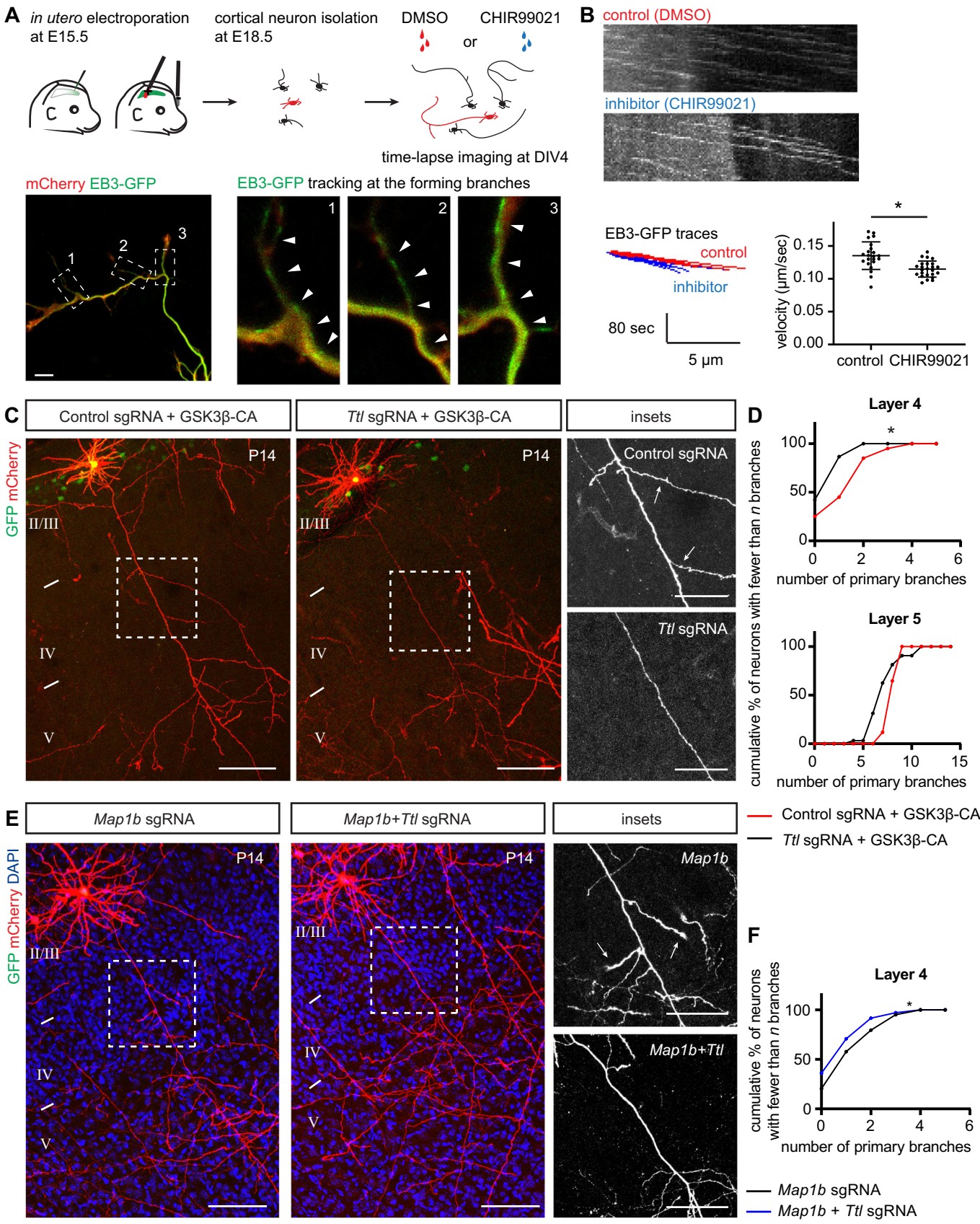

**A** *in utero* electroporation at E15.5 | cortical neuron isolation at E18.5 | DMSO or CHIR99021 | time-lapse imaging at DIV4

mCherry EB3-GFP | EB3-GFP tracking at the forming branches

**B** control (DMSO) | inhibitor (CHIR99021)

EB3-GFP traces — control / inhibitor

80 sec / 5 µm

velocity (µm/sec) — control CHIR99021 *

**C** Control sgRNA + GSK3β-CA | Ttl sgRNA + GSK3β-CA | insets

GFP mCherry | P14 | II/III | IV | V

Control sgRNA | Ttl sgRNA

**D** Layer 4 | Layer 5

cumulative % of neurons with fewer than *n* branches

number of primary branches

— Control sgRNA + GSK3β-CA
— Ttl sgRNA + GSK3β-CA

**E** Map1b sgRNA | Map1b+Ttl sgRNA | insets

GFP mCherry DAPI | P14 | II/III | IV | V

Map1b | Map1b+Ttl

**F** Layer 4

cumulative % of neurons with fewer than *n* branches

number of primary branches

— Map1b sgRNA
— Map1b + Ttl sgRNA

**Figure 6.  GSK3β/MAP1B regulates interstitial axon branching through modification of tubulin tyrosination.**

(A) Schematic of the microtubule dynamics assessment experiment. E15.5 embryos were electroporated with mCherry and EB3-GFP, neurons were isolated at E18.5, and EB3-GFP particles were analyzed at DIV4 using time-lapse microscopy under control conditions (DMSO) or following GSK3β inhibition by CHIR99021. An example image depicts axonal branch sites (insets at the right) where EB3-GFP movements were tracked (arrowheads). See also Movie EV1. Scale bar: 10 μm. (B) Example kymographs from control and experimental conditions, with the traces of individual EB3-GFP particles shown at the bottom. Velocity of EB3-GFP decreases after GSK3β inhibition. Each dot in the velocity plot represents individual neuron (control $n = 22$, GSK3β inhibitor $n = 24$). $*p < 0.05$, $t$-test. (C, D) TTL acts downstream from GSK3β. Removal of *Ttl* via CRISPR/Cas9 in neurons overexpressing GSK3β-CA rescues GSK3β-CA-induced ectopic branching (arrows in insets on right). Data analysis and presentation as in Fig. 1. $*p < 0.05$, $t$-test. Scale bars: 100 μm, 50 μm (insets). (E, F) TTL acts downstream from MAP1B. Removal of *Ttl* via CRISPR/Cas9 in neurons overexpressing GSK3β-CA rescues ectopic interstitial axon branching induced by *Map1b* deletion (arrows in insets on right). Data analysis and presentation as in Fig. 1. $*p < 0.05$, $t$-test. Scale bars: 100 μm, 50 μm (insets). Source data are available online for this figure.

axon branching patterns, and LKB1-NUAK1 kinases regulate terminal CPN axon branching by presynaptic capture of mitochondria (Courchet et al, 2013). Further, Wnk kinases, in both Drosophila mechanosensory neurons and mammalian CPNs, regulate terminal axon branching (Izadifar et al, 2021). However, these signaling molecules do not appear to function in the regulation of interstitial axon branches that arise *de novo* from the axon shaft. Similarly, a wide range of signaling molecules we have assessed elsewhere (Dorskind et al, 2023) also appear dispensable for the regulation of cortical interstitial axon branching. To the best of our knowledge, there are currently very few mouse mutants that display specific defects in cortical interstitial axon branching such as ectopic interstitial ipsilateral axon branching of layer 2/3 CPNs in layer 4 (Dorskind et al, 2023). Further, this stereotypic interstitial branching pattern is likely to arise from the effects of extracellular cues, incorporated within the extracellular matrix or on the surface of cells in layer 4 and 5, that are yet to be discovered.

There are several important considerations when investigating interstitial axon branching. These include distinguishing interstitial axon branching from among the various types of axon branching described to date that include: growth cone splitting, terminal axon branching, and terminal axon branching coupled to synaptogenesis which can be influenced by neural activity (Kalil and Dent, 2014; Dorskind and Kolodkin, 2021). Current labeling techniques are limited with respect to their ability to allow for precise analysis of interstitial axon branching in the developing mammalian brain. The evaluation of cortical axon branching often relies on quantification of fluorescence intensities from axonally expressed fluorophores in large numbers of neurons, which is limited with regard to defining precise levels of interstitial axon branching (Izadifar et al, 2021; De León Reyes et al, 2019; Suárez et al, 2014). Extensive analyses of axon branching in various in vitro systems have been very informative (Tymanskyj et al, 2017; Ketschek et al, 2016; Spillane et al, 2011; Yu et al, 2008). However, the interpretation of results gleaned from dissociated neurons in vitro is complicated since interstitial axon branching in vivo is spatially regulated to allow for laminar-specific innervation by subsets of CPNs, so it is important to complement these studies with in vivo assessments of factors that regulate axon branching.

A central conclusion from this study is that activation of GSK3β can induce interstitial axonal branching. GSK3β has been previously implicated in regulating various aspects of neuronal development, including cortical excitatory neuron migration, neuronal polarization, axon outgrowth and neurotrophin-induced axon branching (Kim et al, 2011). Our bipartite system allowed us to recombine the *Gsk3β* floxed allele in post migratory neurons,

allowing us to expand upon previous work which reported severe cortical neuron migration defects and early postnatal lethality in *Gsk3α⁻/⁻ Gsk3βfl/fl:Neurod6-Cre* mutants (Morgan-Smith et al, 2014). Our data suggest that GSK3β is locally inhibited in layer 2/3 CPN axons within layer 4. In line with previous findings, (Kim et al, 2006; Morgan-Smith et al, 2014) we also observe some redundancy between the GSK3 isoforms α and β—we find that only deletion of both (using *Gsk3α⁻/⁻;Gsk3βfl/fl* mice) inhibits layer 2/3 CPN interstitial axon branching in vivo in layer 5.

We have identified MAP1B, a microtubule-binding protein known to predominantly bind to MT polymer shafts, in our search for downstream GSK3β targets that regulate interstitial axon branching (Bodakuntla et al, 2019). MAP1B regulates neuronal polarity and axon development, but its role in interstitial axon branching has not been quantitatively assessed in vivo (Dajas-Bailador et al, 2012; Meixner et al, 2000; Gonzalez-Billault et al, 2001). However, previous work indicates that MAP1B restricts axon branching in vitro in regenerating DRG axons, (Barnat et al, 2016) and our observations are commensurate with these findings. Interestingly, we found that other GSK3β substrate molecules we investigated in this study, including APC, MACF1 and CLASPs, which are plus-end tracking proteins (+TIPs) and thus bind to the growing end of MTs, (Akhmanova and Hoogenraad, 2005) do not appear to regulate layer 2/3 CPN interstitial axon branching. Given MAP1B binding preferences and our results from EB3-GFP tracking experiments, we speculate that this GSK3β/MAP1B pathway is important for local changes in axon MT dynamics and thus is involved during the initial phase of interstitial axon branch formation (Kalil and Dent, 2014). In addition, we find that GSK3β-mediated phosphorylation of MAP1B is also necessary for collateral axon branch formation since overexpression of dephosphomimetic MAP1B mutant prevented GSK3β-induced axon branching. In agreement with this hypothesis, enrichment of mode I phosphorylated MAP1B has been observed in layer 5 axons in developing mouse cortex, (Majewska and Skangiel-Kramska, 2000) and our immunostaining data also support this observation. Thus, we propose a 'MAP1B brake' model in which a brake on the formation of interstitial axon branches is released by GSK3β-mediated MAP1B phosphorylation.

Previous work shows that MAP1B regulates microtubule dynamics through modulation of tubulin tyrosination cycle; phosphorylated MAP1B causes a nearly complete loss of detyrosinated microtubules (Goold et al, 1999). Tubulin detyrosination, the removal of C-terminal tyrosine from the α-tubulin isoform, can be reversed by tyrosination. The tyrosination/detyrosination cycle is one feature of the tubulin code, which includes multiple post-translational tubulin modifications, each of which is capable of

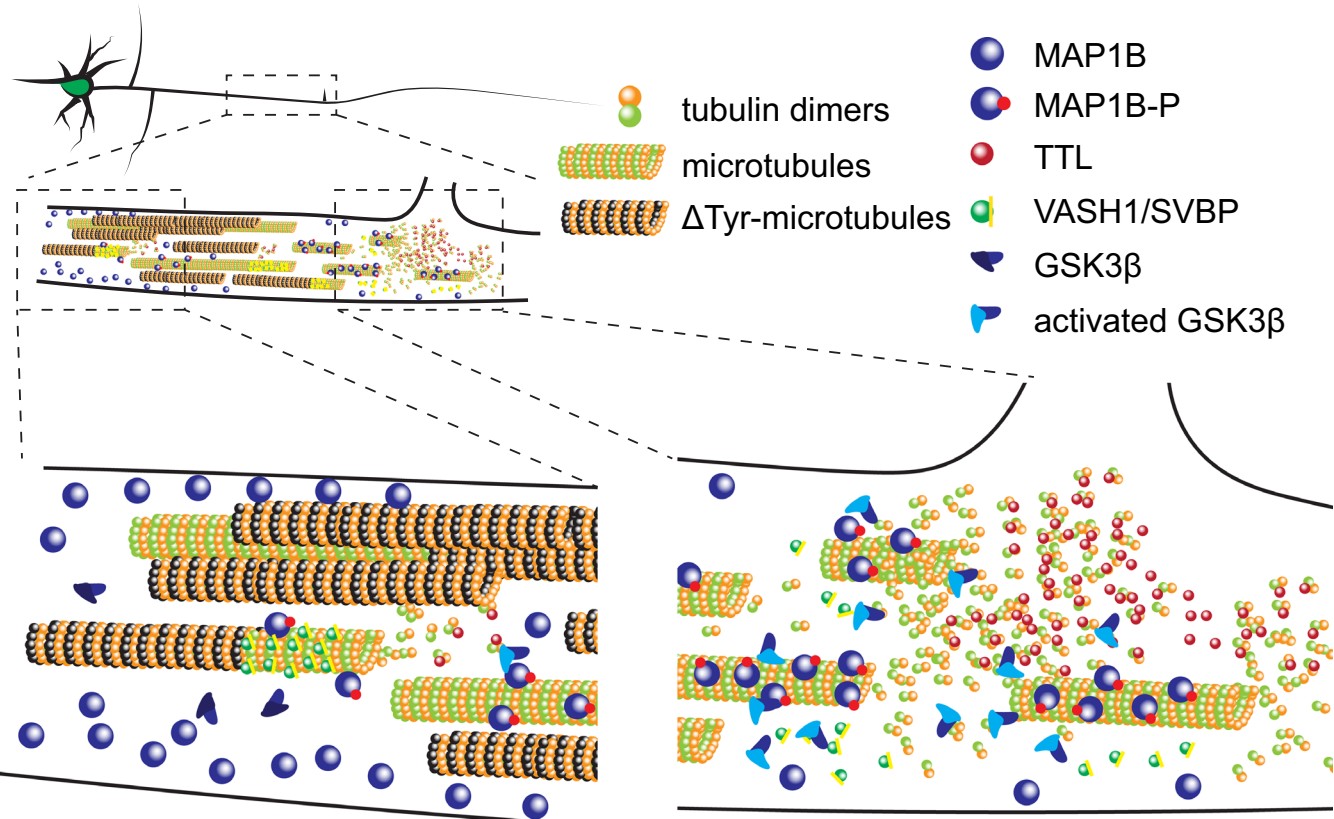

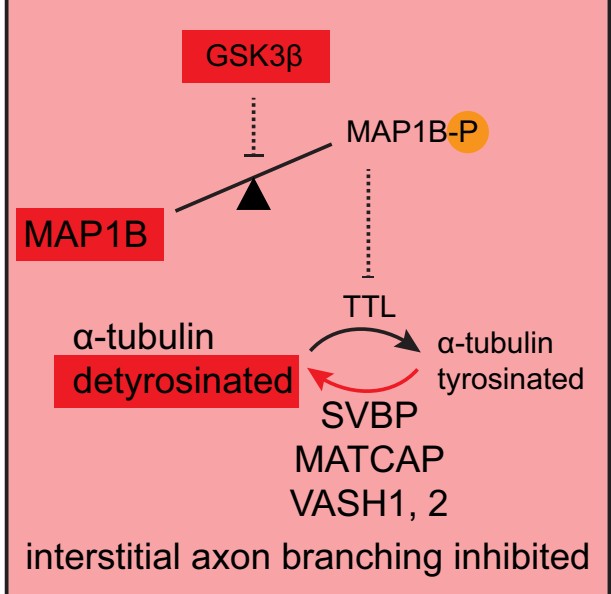

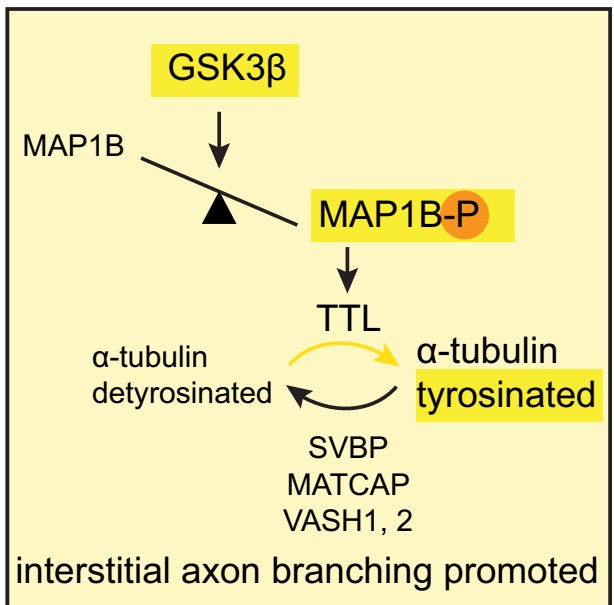

**Figure 7. GSK3β inhibition inhibits microtubule dynamics.**

The intracellular signaling pathway that regulates the formation of interstitial axon branches in cortical neurons. Our results suggest the following model: MAP1B functions as a brake on interstitial axon branching (left, red panel), which can be released following GSK3β-mediated phosphorylation (right, yellow panel). Subsequently, the ratio of tyrosinated/detyrosinated tubulin shifts towards the tyrosinated tubulin, promoting the formation of interstitial axon branches.

exerting distinct effects on cellular functions (Janke and Magiera, 2020; Sanyal et al, 2023). For example, it has been recently shown that the tubulin glutamylases TTLL6 and TTLL11 regulate motor axon development in Zebrafish (Martin et al, 2024). However, the function of the tyrosination cycle in regulating neuronal morphology has not been extensively tested in vivo. Our in vivo loss-of-function and gain-of-function experiments utilizing tyrosination cycle enzymes show that shifting the ratio towards tyrosinated MTs is sufficient to induce interstitial axon branching in layer 2/3 CPNs. *Ttl* loss of function rescues GSK3β-CA-induced ectopic axon branching, indicating that the tyrosinating enzyme TTL acts downstream from GSK3β. Together, our results show that formation of CPN axon collateral branches is regulated by the tubulin code.

How the MAP1B brake and tubulin tyrosination cycle are linked remains to be determined. One study suggests that MAP1B binds TTL independent of phosphorylation (Utreras et al, 2008). Alternatively, the effect of MAP1B on tyrosination cycle could be indirect through modulation of other microtubule posttranslational modifications (Krisenko et al, 2015; Ebberink et al, 2023; Bouquet et al, 2004). Lastly, it is also possible that the regulation happens through the accessibility of the microtubule lattice by detyrosinating enzymes, and that phosphorylated MAP1B protects microtubules from SVBP/VASH binding. We observed that TTL overexpression does not increase branching in neurons overexpressing MAP1B (Appendix Fig. S9). Thus, we hypothesize that GSK3β-mediated phosphorylation of MAP1B could increase its binding to microtubule polymer shafts, as opposed to binding actin. This idea is supported by the observation that dephosphorylated MAP1B binds to actin (Pedrotti and Islam, 1996). Phosphorylated MAP1B would inhibit physical association between MTs and detyrosinating VASH/SVBP complexes. As a result, microtubules would remain tyrosinated, be more dynamic and subject to polymerizing/depolymerizing events more often, and so would lead to axon branch formation. To investigate how tyrosinated MTs affect interstitial axon branching we tested the involvement of the p60 catalytic subunit of Katanin, a MT severing protein known to preferentially act on tyrosinated MTs (Szczesna et al, 2022). However, we did not find evidence that p60 Katanin is sufficient or necessary to modulate CPN interstitial axon branching in vivo (Appendix Figs. S10, S11). This could reflect the need for p60 Katanin at levels we were not able to achieve in our experiments, or functional redundancy among multiple MT severing proteins (including Katanin-like 1 or Spastin). Alternatively, it is possible that other tubulin posttranslational modifications cold modulate the extent of detyrosination (Ebberink et al, 2023) or Katanin's preference for tyrosinated microtubules (Szczesna et al, 2022).

Our results uncover an intracellular signaling pathway that cell-autonomously regulates interstitial axon branching in CPNs during neural development. However, the stereotypic laminar-specific patterning of interstitial axon branches in the neocortex likely reflects unknown instructive and/or permissive cues that act locally in cortical layers 4 and 5. The activity of GSK3β is tightly controlled by Protein Kinase B and PTEN, (Beurel et al, 2015) and so it is possible that these enzymes serve as a link between a cell surface receptor and GSK3β. The identity of cell surface proteins capable of transmitting laminar-specific extracellular signals that regulate GSK3β/MAP1B/MT tyrosination-mediated interstitial axon branching is an important next step in understanding how cortical circuits are established during development.

In conclusion, using methodologies that allow for robust, sparse labeling and precise spatiotemporal genetic manipulation of layer 2/3 CPNs we have identified molecular regulators of interstitial axon branching. We propose a model whereby a MAP1B brake restricts interstitial axon branching until it is released by GSK3β phosphorylation, allowing for regulated generation of a pool of tyrosinated microtubules in the axonal shaft. These observations are an important first step in understanding the generation of stereotypical laminar-specific CPN axon branches and may be generally applicable to multiple populations of cortical excitatory projection neurons.

## Methods

### Animals

Experiments were carried out in strict accordance with the recommendations in the Guide for the Care and Use of Laboratory Animals of the NIH. The animal protocol was approved by the Animal Care and Use Committees of the Johns Hopkins University School of Medicine (Protocol #:MO23M68). Timed pregnant CD1 wild-type females were obtained from Charles River Laboratory. The day of birth was designated as P0. Mice of both sexes were used in all experiments and were housed in a 12:12 light-dark (LD) cycle. $Gsk3\alpha^{-/-}$ and $Gsk3\beta^{fl/+}$ mutant mice were gifts from Feng-Quan Zhou, (Morgan-Smith et al, 2014) $Apc^{fl/fl}$ mice were gift from dr. Bart Williams (Bruxvoort et al, 2007) and $\beta$-catenin$^{fl/fl}$ mice were gift from Jeremy Nathans (Brault et al, 2001).

### Plasmids

All plasmids used in this study are listed in the Table EV2 and are available upon request. Plasmids used for the bipartite labeling method are as described (Dorskind et al, 2023). Supernova plasmids are as described (Luo et al, 2016). Plasmids used for knockdown experiments are as described (Stegmeier et al, 2005). DNA encoding MAP1B was amplified from a GFP-MAP1B-expressing plasmid (a gift from Phillip Gordon-Weeks, Addgene# 44396 (Scales et al, 2009)), the N-terminal GFP sequence was removed and the DNA sequence encoding the Flag epitope was added in frame at the C-terminus. In addition, we introduced a silent mutation to create an AgeI restriction site, simplifying the cloning of phospho-mutant constructs. These were generated by PCR mutagenesis of the core portion of MAP1B (between the endogenous HindIII site and a new AgeI site) and included phosphomimetic (S1260D, T1265D, S1391D) and dephosphomimetic (S1260A, T1265A, S1391A) mutations. Coding DNA sequences for GSK3β, TTL, SVBP, VASH1 and VASH2 were amplified from mouse cortical P5 cDNA. The TTL-DN plasmid was generated by PCR mutagenesis and contained K198A and D200A mutations (Szyk et al, 2011). EB3-GFP DNA sequence was a gift from Niels Galjart&Anna Akhmanova (Stepanova et al, 2003) (Addgene# 190164) and cloned into pCAG plasmid. Sequences of TKIT donor plasmids were amplified from mouse genomic DNA and GFP with a short linker placed in frame at the 5' or 3' end, in line with the amino-terminal or carboxy-terminal tagging.

Polycistronic CRISPR plasmids pX330 (Cong et al, 2013) and pX458 (Ran et al, 2013) were optimized so that one vector contained two U6 promotors with two sgRNA sequences (Fang et al, 2021). Guide RNA sequences were chosen as described (Doench et al, 2016; Morgens et al, 2017). The DNA sequence encoding the TagRFP-T_A1aY1 Tyr-T sensor was amplified from Addgene plasmid #158751 (a gift from Minhajuddin Sirajuddin (Kesarwani et al, 2020)) and cloned into the pMini donor plasmid together with the CAG promoter and bGH polyA signal. In all cases, multiple fragments were assembled together using a HiFi NEBuilder kit. All new constructs were verified by restriction reaction and sequencing. For the purpose of in utero electroporations, endotoxin-free plasmid DNA was prepared using Nucleo-Bond® Xtra Midi EF kit.

## Surgeries

In utero electroporations were performed as described (Hand et al, 2015). Briefly, timed pregnant E15.5 mice were anesthetized with isoflurane and placed on a heating pad. The incision site was shaved, cleaned, and local anesthetic (Bupivacain) was applied subcutaneously. A small laparotomy was performed, and the embryos were removed and rinsed with warm sterile PBS. The lateral ventricles were then injected with small volumes (approximately 0.5–1 µl) of endotoxin-free DNA solutions in PBS with Fast Green dye. Embryos were electroplated with gene paddles (Harvard Apparatus) using a BTX square pulse electroporator (parameters: 40 V, three 50-ms pulses with a 950-ms interval). Animals were placed on a heating pad to recover and pain was further controlled by subcutaneous injection of Buprenorphine. All procedures were conducted in accordance to our IUCAC-approved protocol.

## Tamoxifen injections

10× stock solution was prepared as follows: Tamoxifen (10 mg) was dissolved in 250 µl Ethanol and the solution was mixed with 750 µl Corn Oil. Stock solution was stored at 4 °C up to 10 days. 1× tamoxifen solution was prepared by dissolving the stock solution in Corn Oil, resulting in 1 mg/ml final concentration. Approximately 30 µl of 1× Tamoxifen was applied intragastrically (milk spot injection) at P1 and P2.

## Brain processing and immunostaining

Antibodies used in this study are listed in Table EV2. Animals were transcardially perfused with a modified PHEM buffer (27 mM PIPES, 25 mM HEPES, 5 mM EGTA, 0.47 mM $MgCl_2$, pH = 6.9) followed by ice-cold 4% PFA/PHEM with 5% sucrose and 0.1% Triton-X100. Brains were isolated and post-fixed overnight at 4 °C. Then, all samples were washed multiple times in PBS, sliced coronally (250 µm sections) and incubated in a blocking buffer for at least 30 min at the room temperature (3% BSA with 0.3% Triton-X100 and 0.02% Sodium Azide in PBS, filtered). Brain sections were incubated with primary antibodies dissolved in a blocking buffer with 5% normal goat serum (NGS) overnight at 4 °C. Slices were then washed three times (each round at least 1 h, room temperature) in PBST (PBS with 0.1% Triton-X100) and incubated in secondary antibodies and DAPI dissolved in PBST overnight at

4 °C. Following another three washes, slices were mounted on a glass slides using a Fluorogel/DABCO.

Analysis of tyrosination levels in neurons after genetic manipulations of the tubulin tyrosination cycle was performed as follows: mice (at P4–P7) were perfused and brains isolated as described before. 100 µm thick slices were prepared and immunostained as described above. We used a rat YL1/2 antibody that specifically recognizes tyrosinated MTs. Tyrosination fluorescence signals were measured in ImageJ in GFP+ (electroporated) neurons and compared to fluorescence of neighboring GFP-negative neurons from the same slice.

Analysis of tubulin tyrosination in individual axons expressing the TagRFP sensor (Fig. 4A) was performed as follows: mice (P4) were perfused and brains isolated as described before. 100 µm thick slices were prepared and immunostained as described above with antibodies directed against GFP, RFP, and α-tubulin. The sensor's fluorescence was measured in ImageJ: we traced the axon passing through layer 4 and layer 5 and along the same trace measured the ratio of the RFP/α-tubulin. Layer 5 and layer 4 ratios were compared using a Wilcoxon Matched-pairs test.

## Neuronal culture

Cortical neurons were cultured as described, (Dorskind et al, 2023) with slight modifications. In brief, E15.5 embryos were in utero electroporated with plasmid expressing pCAG-mCherry and pCAG-EB3-GFP. At E18.5, electroporated cortices were dissected into cold HBSS (Thermo Fisher, without $Mg^{2+}$ and $Ca^{2+}$) with 10 mM HEPES (pH 7.3) and digested 20 min in Digestion Solution (4.2 mM $NaHCO_3$, 25 mM HEPES, 137 mM NaCl, 5 mM KCl, 7 mM $Na_2HPO_4$) with DNase I (DN25, Sigma, final concentration 1.44 mg/ml) and papain (P-4762, Sigma, final concentration 1 mg/ml) at 37 °C. Next, neurons were rinsed in HBSS and Neurobasal supplemented with 2.5% B27 (Gibco), 2.5 mM glutamax (Thermo Fisher) and 1% Penicillin/Streptomycin (Thermo Fisher, 100 units/ml Penicillin, 100 µg/ml Streptomycin). Cortices were triturated and cells filtered with 40-µm cell strainer. Neurons were plated on a glass-bottom cell culture plates (100,000 cells/well) coated with Poly-D-Lysine (50 µg/ml) and Laminin (5 µg/ml). At DIV4, EB3-GFP was tracked under acute GSK3β pharmacological inhibition (using 10 µM CHIR99021 inhibitor, dissolved in DMSO (Bennett et al, 2002)) as follows: culture plate was transferred into the Zeiss LSM confocal microscope equipped with heated chamber and 5% $CO_2$ atmosphere, and neurons were imaged with 63x objective at 4 s intervals.

## Microscopy

Brain slices were imaged with inverted or upright Zeiss 700 confocal microscopes using a 20× objective (Plan-Apochromat 20×/0.8). Imaging parameters were as follows: px size 0.31 × 0.31 µm (2048 × 2048 px resolution at 0.5 zoom, total image size was 640 × 640 µm), optical sections 1.8 µm, z-stack 1.5 µm. 50–70 z-stack images were taken per neuron, which usually covered the main axon extending from the cell body to the border between layers 5 and 6. Dendritic spines shown in Fig. EV3B and EB3-GFP particles shown in Fig. 6A and Movie EV1 were captured using 63× oil-immersion objective (Plan-Apochromat 63×/1.4 Oil DIC).

## Quantification and statistical analysis

Confocal image stacks were manually analyzed in ImageJ. First, we defined the borders between layers 4 and 5 according to the DAPI signal. Primary interstitial axon branches elaborated from the main axon were quantified in the layer 4 and 5. For neurons in which the main axon in layer 5 was missing (due to sectioning) only layer 4 branches were counted, and vice versa. Orientation of basal dendrites (Fig. EV1G) was measured as follows: a line at the base of analyzed neuron was drawn (in right angles to the apicobasal neuron orientation) and the number of dendritic intersections with the line was quantified. Time-lapse images of EB3-GFP particles were analyzed by KymoToolBox plugin (Zala et al, 2013). The numbers of analyzed neurons/animals are listed in the Table EV1. Branching data were statistically analyzed by fitting a mixed effects model (Maxwell et al, 2018) as implemented in GraphPad Prism 9.5.0. This mixed model uses a compound symmetry covariance matrix and is fit using Restricted (Residual) Maximum Likelihood (REML). Significance was determined with either a *t*-test or a one-way ANOVA with a post-hoc Dunnet's test. Distribution of tyrosination signal in axons (Fig. 4B) was analyzed with Wilcoxon matched-pairs test. All p values are reported in Table EV1.

## Data availability

Source data are provided with this paper. A list of reagents used in the study is provided in Table EV2. All raw microscopy data and analyses related to this paper will be deposited at the Johns Hopkins Research Data Repository that is managed by Johns Hopkins Data Services (JHDS). This repository provides public access to these data through an established platform supported by storage and preservation practices that follow the Open Archival Information System reference model. Deposited data have been given a standard data citation and persistence identifier (DOI): https://doi.org/10.7281/T1/NSHJC5. Data are archived under a memorandum of understanding and renewed every 5 years with the PI's consent. This study includes no data deposited in external repositories.

## Peer review information

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

## Acknowledgements

We thank members of the Kolodkin laboratory for thoughtful input on this work. We thank Ulrich Mueller, Feng-Quan Zhou and Takanari Inoue for comments on the manuscript, and Jeremy Nathans, Feng-Quan Zhou and Bart Williams for mouse reagents. We also thank Nicholas M. Kanaan for providing us with nonphosphorylated GSK3β antibody and Jay Vaidya for consultation on statistics. This work was supported in part by the Howard Hughes Medical Institute (ALK), the CMM Graduate Training Program at The Johns Hopkins University School of Medicine–T32-GM008752 (JD), the Kavli Neuroscience Discovery Institute at Johns Hopkins (SS), and an EMBO Postdoctoral Fellowship no. 364-2021 (JZ).

## Author contributions

**Jakub Ziak**: Conceptualization; Resources; Data curation; Formal analysis; Funding acquisition; Investigation; Writing—original draft; Writing—review and editing. **Joelle Dorskind**: Resources; Funding acquisition; Investigation. **Brian Trigg**: Resources; Investigation. **Sriram Sudarsanam**: Resources; Funding acquisition. **Xinyu O Jin**: Resources. **Randal Hand**: Resources. **Alex L Kolodkin**: Supervision; Funding acquisition; Writing—original draft; Project administration; Writing—review and editing.

## Disclosure and competing interests statement

The authors declare no competing interests. RH currently serves as a Director of Neuroscience at Prilenia Therapeutics. JD currently is a postdoctoral researcher at Novartis.

# Expanded View Figures

**Figure EV1.  GSK3β regulates neuronal morphology in layer 2/3 CPNs (related to Fig. 1).**

(A) Overview of original candidate molecule screen (Dorskind et al, 2023) for identification of CPN interstitial axon branching regulators. (B) Overexpression of human constitutively active GSK3β (GSK3β-CA) induces ectopic interstitial axon branching in layer 2/3 CPNs. Scale bars: 100 μm. (C) Additional changes in axonal morphology induced by human GSK3β-CA include axonal loops and spiny protrusions along axons (arrowheads). Scale bars: 100 μm (including callosal loop and protrusions, right), 50 μm (primary axon loop, bottom right). (D) GSK3β-CA-induced interstitial axon branching (arrows) in layer 2/3 CPNs persist until adulthood. Scale bars: 100 μm. (E) Example of a severe loss of interstitial axon branches following removal of both *Gsk3α* and *Gsk3*β isoforms in the mouse (related to Fig. 1D). Scale bars: 100 μm. (F) Quantification of axon interstitial branching in $GSK3α^{-/-};β^{+/+}$ mutant mice. Controls are the same as those shown in Fig. 1D. (G) Basal dendrite patterning in layer 2/3 CPNs is regulated by GSK3 isoforms. Red dotted line (placed at right angle to the apicobasal orientation of a neuron) highlights the region where dendritic intersections were calculated. Scale bars: 50 μm. (H) Quantification of dendritic intersections from (G). Each column represents a single animal; mean ± SD is plotted. $GSK3α^{+/-};β^{fl/fl}$ $n = 3$ mice, $GSK3α^{-/-};β^{fl/fl}$ $n = 4$ mice, $^*p < 0.05$, $t$-test. Source data are available online for this figure.

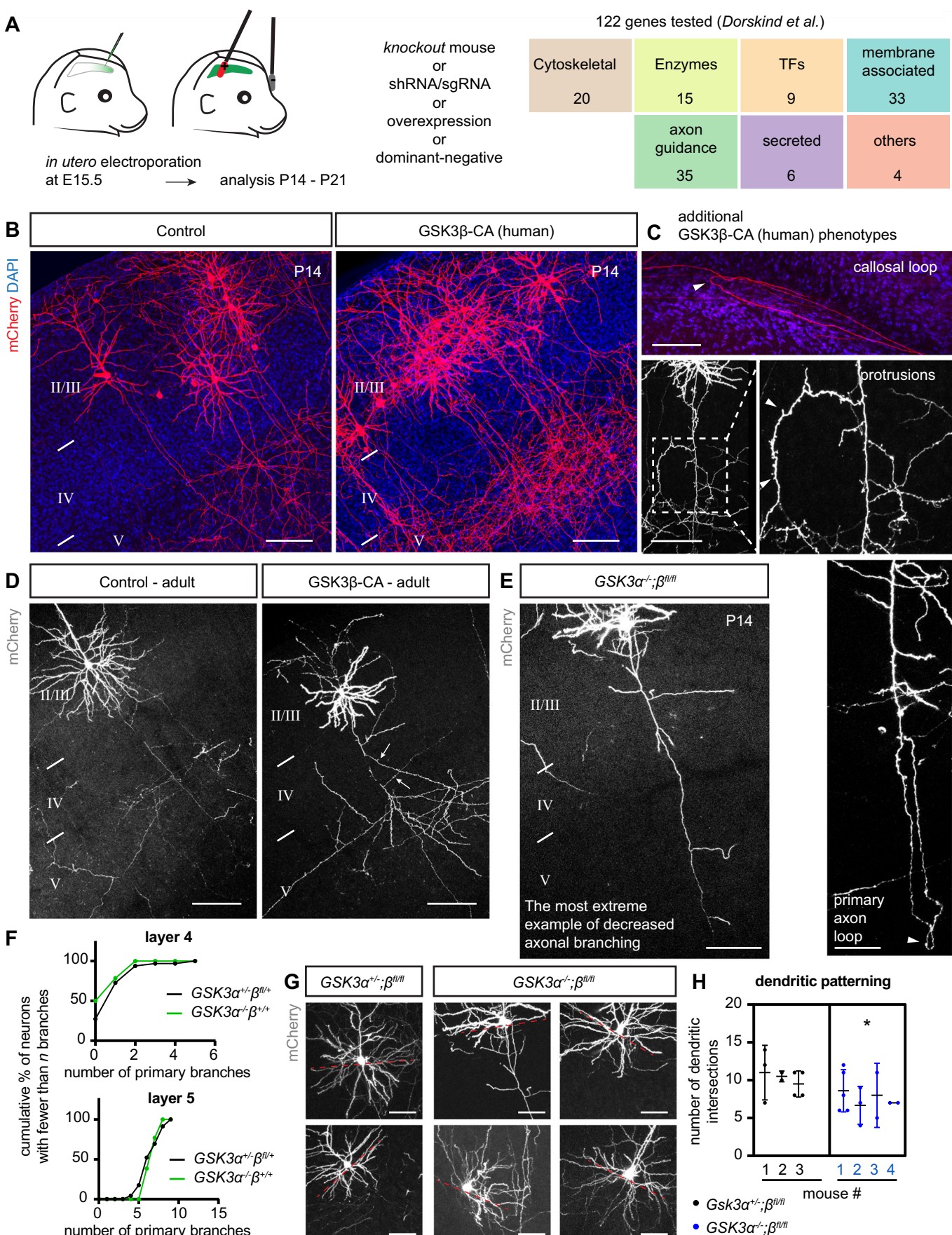

A

in utero electroporation
at E15.5 → analysis P14 - P21

122 genes tested (*Dorskind et al.*)

*knockout* mouse
or
shRNA/sgRNA
or
overexpression
or
dominant-negative

| Cytoskeletal | Enzymes | TFs | membrane associated |
|---|---|---|---|
| 20 | 15 | 9 | 33 |

| axon guidance | secreted | others |
|---|---|---|
| 35 | 6 | 4 |

B

mCherry DAPI

Control — P14

GSK3β-CA (human) — P14

II/III
IV
V

C

additional
GSK3β-CA (human) phenotypes

callosal loop

protrusions

D

mCherry

Control - adult

GSK3β-CA - adult

II/III
IV
V

E

GSK3α⁻/⁻;β^{fl/fl}

mCherry

P14

II/III
IV
V

The most extreme
example of decreased
axonal branching

primary
axon
loop

F

layer 4

*GSK3α^{+/-}β^{fl/+}*
*GSK3α^{-/-}β^{+/+}*

cumulative % of neurons
with fewer than *n* branches

number of primary branches

layer 5

*GSK3α^{+/-}β^{fl/+}*
*GSK3α^{-/-}β^{+/+}*

number of primary branches

G

mCherry

*GSK3α^{+/-};β^{fl/fl}*

*GSK3α^{-/-};β^{fl/fl}*

H

dendritic patterning

number of dendritic
intersections

*

mouse #

● *Gsk3α^{+/-};β^{fl/fl}*
● *GSK3α^{-/-};β^{fl/fl}*

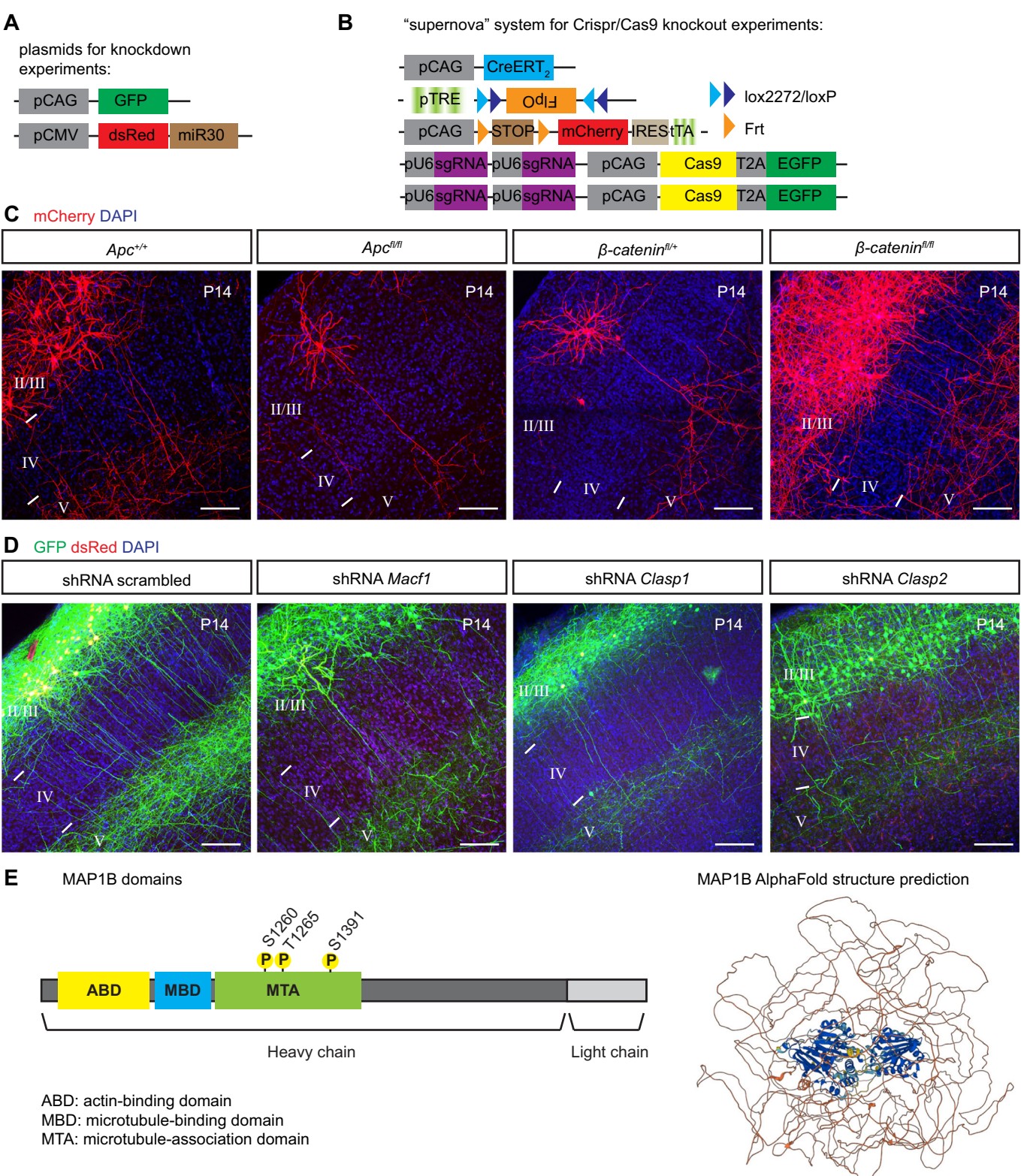

**A** plasmids for knockdown experiments:

**B** "supernova" system for Crispr/Cas9 knockout experiments:

**C** mCherry DAPI

$Apc^{+/+}$ | $Apc^{fl/fl}$ | $\beta$-catenin$^{fl/+}$ | $\beta$-catenin$^{fl/fl}$

**D** GFP dsRed DAPI

shRNA scrambled | shRNA *Macf1* | shRNA *Clasp1* | shRNA *Clasp2*

**E** MAP1B domains

S1260 T1265 S1391

ABD MBD MTA

Heavy chain | Light chain

ABD: actin-binding domain
MBD: microtubule-binding domain
MTA: microtubule-association domain

MAP1B AlphaFold structure prediction

**Figure EV2.  Candidate survey for GSK3β targets that regulate interstitial axon branching (related to Fig. 2).**

(A) Experimental approach for shRNA-mediated knockdown experiments. (B) Experimental approach for CRISPR/Cas9-mediated knockout experiments. (C) Removal of *Apc* or β-*catenin* using conditional knockout mice does not influence interstitial axon branching in layer 4. Scale bars: 100 μm. (D) Knockdown of *Macf1, Clasp1* or *Clasp2* does not influence interstitial axon branching in layer 4. Scale bars: 100 μm. (E) Domain structure of MAP1B protein and an AlphaFold structure prediction model (AF-P14873-F1). Note that the confidence provided by the AlphaFold algorithm model is very low for MAP1B. Source data are available online for this figure.

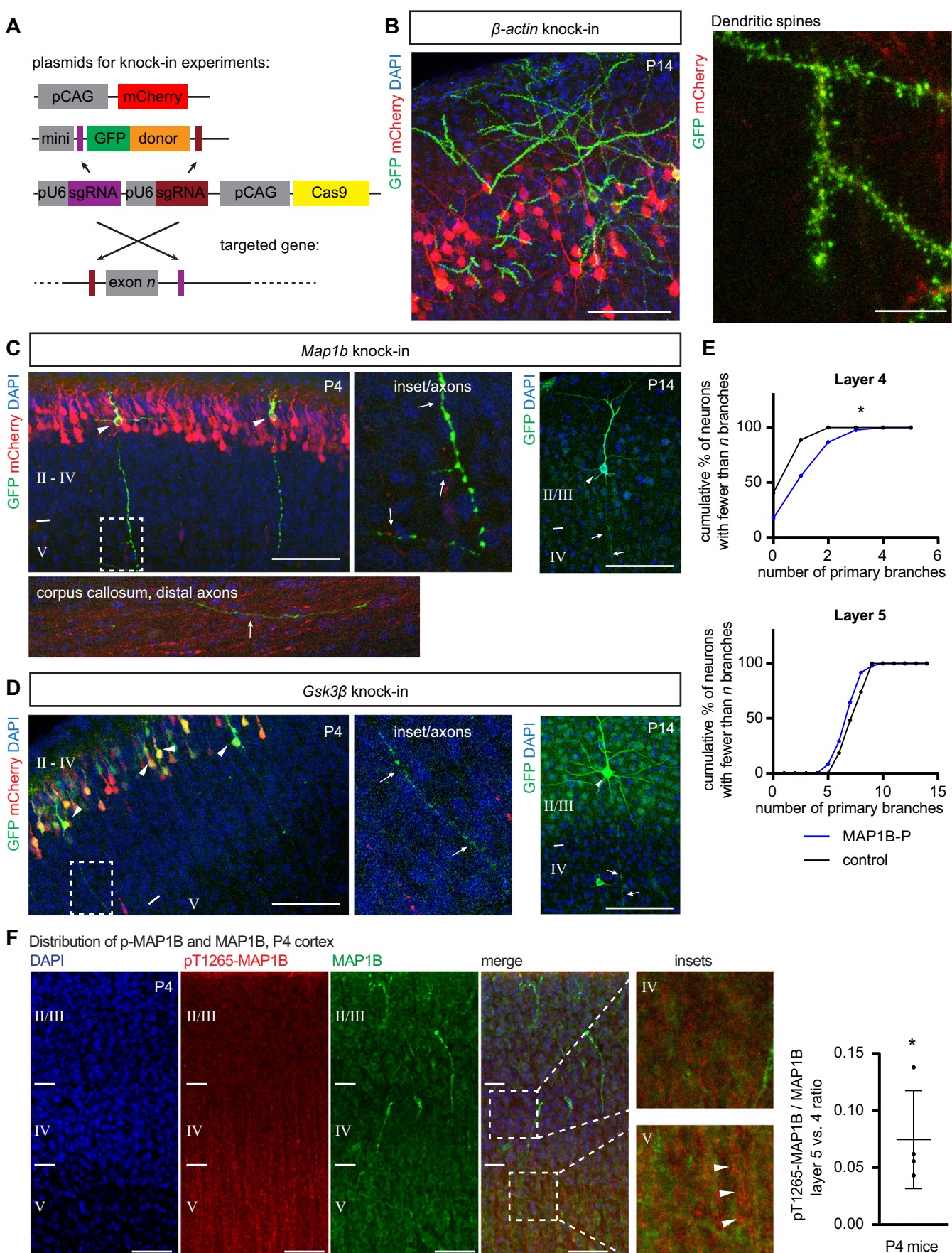

◀ **Figure EV3. Analysis of the GSK3β target MAP1B subcellular localization in cortical neurons using CRISPR/Cas9 endogenous tagging (related to Figs. 2 and 3).**

(**A**) Experimental design for endogenous tagging experiments using CRISPR/Cas9 knock-in approach. (**B**) Example of endogenous GFP-β-actin tagging. Note accumulation of GFP in actin-rich dendritic spines (dendritic spines, right). Scale bars: 100 μm (left), 10 μm (dendritic spines). (**C, D**) Representative images showing endogenous tagging of *Map1b* and *Gsk3*β at P4 or P14 cortices (arrowheads and arrows highlight expression in somatodendritic compartments and axons, respectively). Note strong expression of MAP1B, which fills entire neuron, including the most distal axonal compartments in the corpus callosum (panel **C**, bottom). Insets in C and D show axonal expression. Scale bars: 100 μm. (**E**) Overexpression of MAP1B-P leads to ectopic interstitial axon branching in layer 4. For this analysis, control neurons from Fig. 2E were used. (**F**) Phosphorylation of MAP1B in the cortex. Cortical slices from P4 mice were probed with phopsho-MAP1B (pT1265-MAP1B, red) and MAP1B (green) antibodies. Phosphorylated MAP1B is enriched in cortical layer 5. Note the punctate pattern in bundled structures in layer 5 (insets, arrowheads). Right: pT1265-MAP1B/MAP1B intensity in the cortical layer 4 and layer 5 was compared using a Paired t test and is presented as a layer 5/layer 4 ratio (individual values from $n = 4$ mice, together with mean ± S.D. are plotted). *$p < 0.05$, t-test. Scale bar: 50 μm. Source data are available online for this figure.

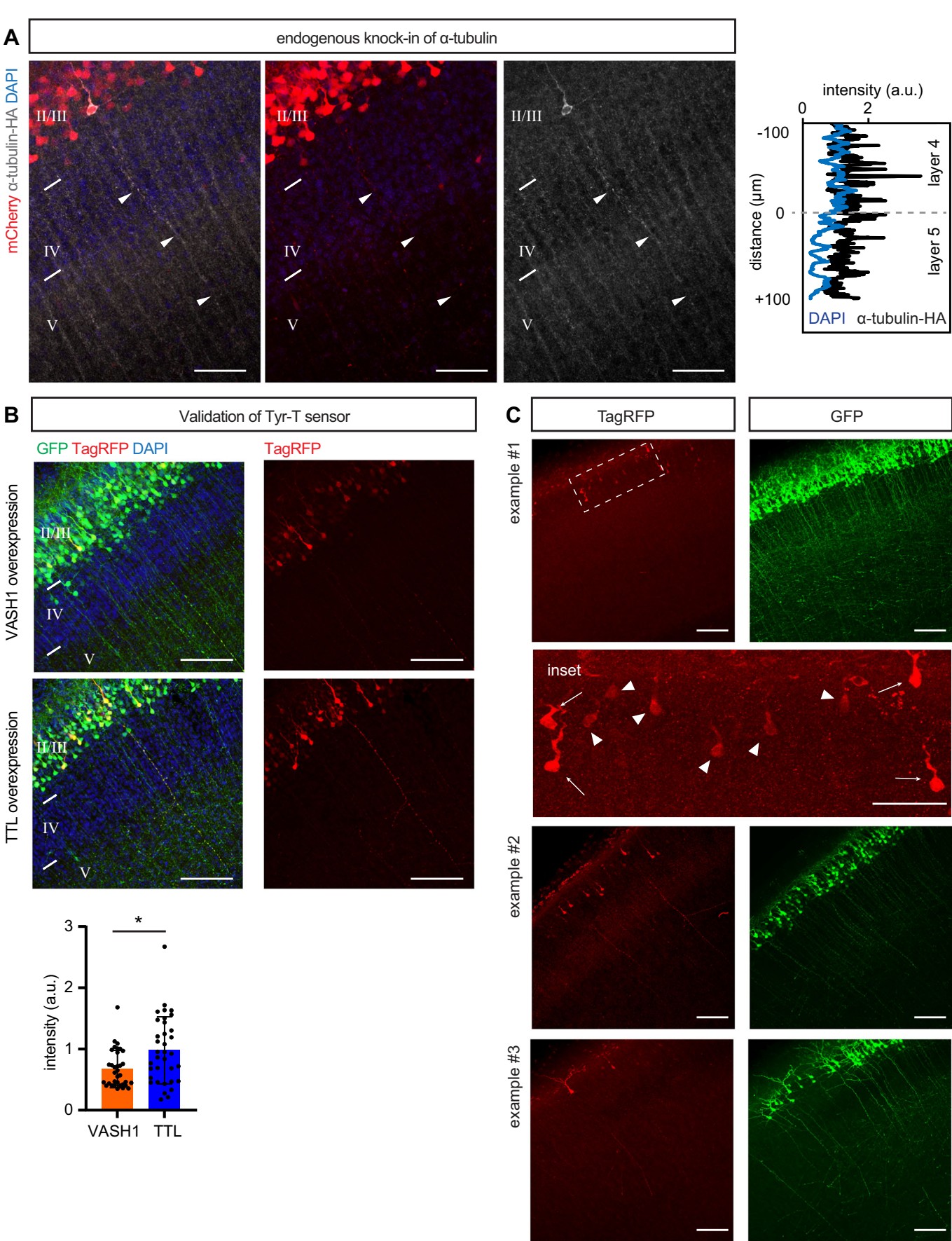

◀ **Figure EV4.  Distribution of total and tyrosinated α-tubulin in S1 neocortex (related to Fig. 4).**

(A) Example of endogenous α-tubulin-HA tagging (exon 1). Note equal distribution of HA within the axon (arrowheads). Sample trace on the right demonstrates fluorescence intensity of DAPI (a measure of cortical layer) and HA. Scale bars: 50 μm. (B) Validation of the Tyr-Tubulin sensor. Neurons were electroporated with the sensor plasmids together with TTL-IRES-GFP or VASH1-IRES-GFP plasmids. Overexpression of VASH1 decreases the fluorescence of the sensor in comparison with TTL, demonstrating its' feasibility. Data shown at the bottom of this panel are presented as mean ± st.d. TTL $n = 34$ neurons, VASH1 $n = 35$ neurons *$p < 0.05$, $t$-test. Scale bars: 100 μm. (C) Examples of a fluorescence intensity distributions after targeting *Rosa26* locus via the TKIT approach (see main text). In the detail for the example #1, note that a small population of neurons expresses high levels of TagRFP (arrows), while a larger population of neurons expresses lower levels of TagRFP (arrowheads). Scale bar: 100 μm, 50 μm (inset). Source data are available online for this figure.

                                                                    *The EMBO Journal*  Volume 43 | Issue 7 | April 2024 | 1214–1243          **EV8**

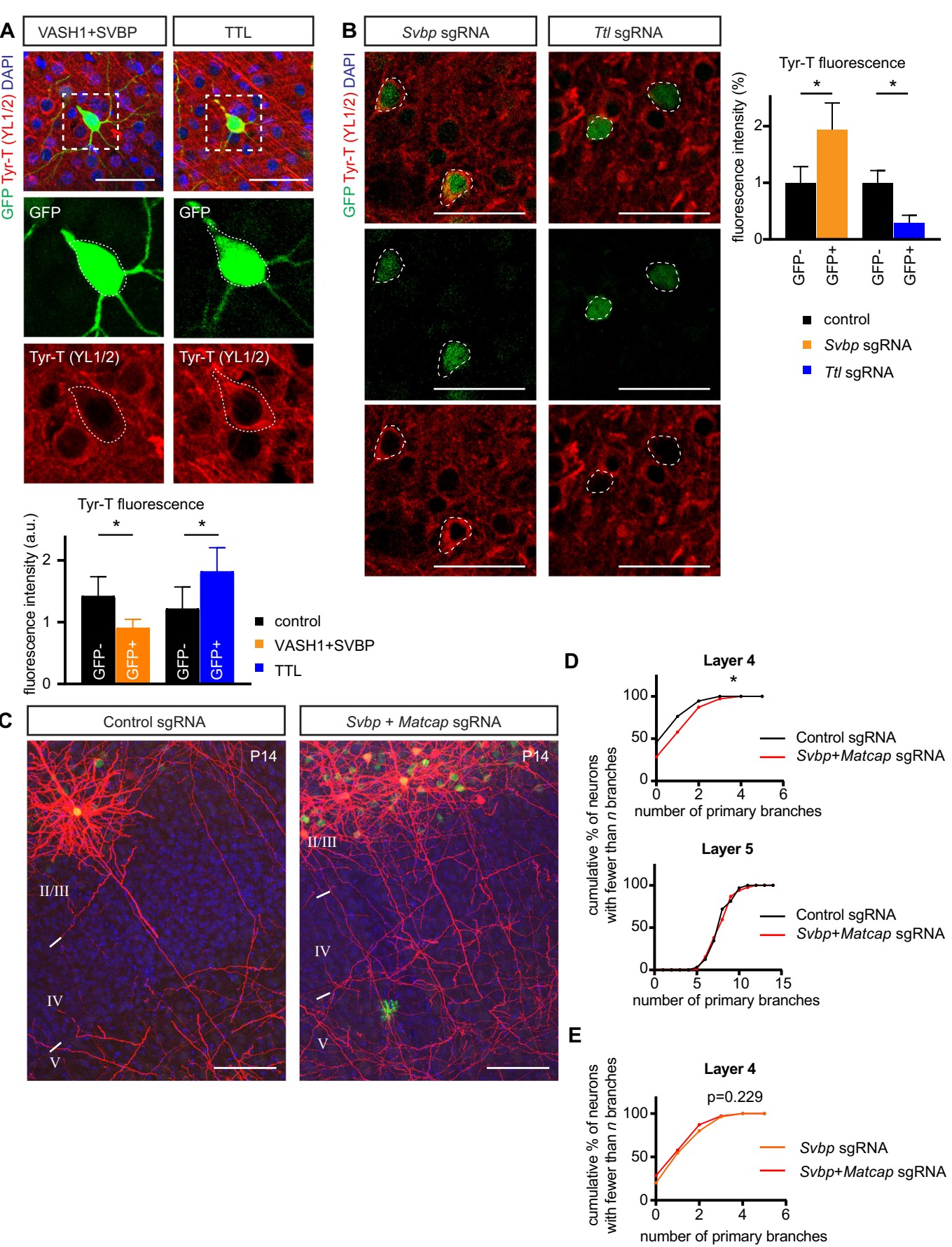

◀  **Figure EV5.  Tyrosination of α-tubulin in S1 neocortex under normal and experimental conditions (related to Figs. 4 and 5).**

(**A**) Immunostaining of Tyr-T after TTL and VASH1/SVBP overexpression experiments. Note lack of a tyrosination signal following VASH1/SVBP overexpression and an increase in the tyrosination signal after TTL overexpression. Data shown at the bottom are presented as mean ± st.d. TTL control $n = 10$ neurons, TTL overexpression $n = 3$ neurons, SVBP control $n = 12$ neurons, SVBP overexpression $n = 4$ neurons. *$p < 0.05$, $t$-test. Scale bar: 50 μm. (**B**) Immunostaining of Tyr-T following CRISPR knockdown. Note decrease of the tyrosination signal after TTL deletion and an increase in the signal after SVBP deletion. GFP labels electroporated cells that harbor the pX458 plasmid (cells with a dashed contour). Quantification of the tyrosination signal in targeted (GFP$^+$) cells, normalized to GFP$^-$ cells from the same slices, is shown at the bottom. Data are presented as mean ± st.d. TTL control $n = 18$ neurons, TTL deletion $n = 21$ neurons, SVBP control $n = 21$ neurons, SVBP deletion $n = 22$ neurons. *$p < 0.05$, $t$-test. Scale bar: 50 μm. (**C**, **D**) Simultaneous removal of *Svbp* and *Matcap* promotes interstitial axon branching to the same extent as *Svbp* deletion alone. Data analysis and presentation are as in Fig. 1. *$p < 0.05$, $t$-test. Scale bars: 100 μm. (**E**) Comparison of *Svbp* knockdown with combined *Svbp*+*Matcap* knockdown. $t$-test. Scale bars: 100 μm. Source data are available online for this figure.

