## [Peer Review File · The EMBO Journal]

Microtubule-binding protein MAP1B Regulates Interstitial Axon Branching of Cortical Neurons via the Tubulin Tyrosination Cycle

Jakub Ziak, Joelle Dorskind, Brian Trigg, Sriram Sudarsanam, Xinyu Jin, Randal Hand, and Alex L. Kolodkin

Corresponding author: Alex L. Kolodkin (kolodkin@jhmi.edu)

Review Timeline:

Submission Date:	11th Aug 23
Editorial Decision:	22nd Sep 23
Revision Received:	15th Dec 23
Editorial Decision:	23rd Jan 24
Revision Received:	24th Jan 24
Accepted:	25th Jan 24

Editor: Ieva Gailite

Transaction Report:

Dear Alex,

Thank you for submitting your manuscript for consideration by the EMBO Journal. We have now received comments from three reviewers, which are included below for your information.

As you will see from the reports, all reviewers find the study of interest, while also pointing out a number of important aspects that would need to be strengthened in the final manuscript before they can recommend acceptance of the manuscript. In particular, reviewers #1 and #3 find that direct evidence for MAP1B phosphorylation by GSK3beta would need to be provided. They further indicate that the relationships between MAP1B, tubulin tyrosination and microtubule dynamics need further clarification. Finally, reviewers #2 and #3 point out that a better quantification of tubulin tyrosination would be required.

Based on the interest expressed in the reports, I would like to invite you to address the issues raised by the referees in a revised manuscript. I think it would be useful to discuss the revision in more detail via email or phone/videoconferencing - please let me know which option you prefer.

We generally allow three months as standard revision time, which can be extended to six months in the case of major revisions. As a matter of policy, competing manuscripts published during this period will not negatively impact on our assessment of the conceptual advance presented by your study. However, please contact me as soon as possible upon publication of any related work to discuss the appropriate course of action. Should you foresee a problem in meeting this deadline, please let us know in advance to discuss an extension.

When preparing your letter of response to the referees' comments, please bear in mind that this will form part of the Review Process File and will therefore be available online to the community. For more details on our Transparent Editorial Process, please visit our website: <https://www.embopress.org/page/journal/14602075/authorguide#transparentprocess>. Please also see the attached instructions for further guidelines on preparation of the revised manuscript.

Please feel free to contact me if you have any further questions regarding the revision. Thank you for the opportunity to consider your work for publication. I look forward to discussing your revision.

With best regards,

Ieva

- a point-by-point response to the referees' comments, with a detailed description of the changes made (as a word file).
- a word file of the manuscript text.

- individual production quality figure files (one file per figure)
- a complete author checklist, which you can download from our author guidelines (<https://www.embopress.org/page/journal/14602075/authorguide>).
- Expanded View files (replacing Supplementary Information)
Please see out instructions to authors
<https://www.embopress.org/page/journal/14602075/authorguide#expandedview>

We realize that it is difficult to revise to a specific deadline. In the interest of protecting the conceptual advance provided by the work, we recommend a revision within 3 months (21st Dec 2023). Please discuss the revision progress ahead of this time with the editor if you require more time to complete the revisions.

Referee #1:

Ziak et al investigate interstitial axon branching in layer 2/3 callosal projection neurons (CPNs) within the mammalian brain. The study employs a unique approach that enables precise labeling of these neurons, allowing for both quantitative analysis of axonal morphology and targeted gene expression manipulation. The study's central finding is that the serine/threonine kinase GSK3 β plays a pivotal role in promoting interstitial axon branching in layer 2/3 CPNs. Activation of GSK3 β triggers a cascade of events that leads to the release of MAP1B-mediated inhibition of axon branching. This molecular pathway is proposed to serve as a "brake" on axon branching, which can be released by GSK3 β activation. Furthermore, the study identifies that GSK3 β /MAP1B signaling influences the tyrosination/detyrosination cycle of α -tubulin, which is proposed to be part of the underlying mechanism of interstitial axon branch regulation.

Technically and analytically, the paper is strong with excellent imaging, great genetic tools, sound analysis and statistics. The data are clear. Conceptually, however, the links between GSK3beta, MAP1B, tyrosination/detyrosination and interstitial branching are tenuous, and the overall conclusions/models proposed are based on artificial manipulations with very little evidence with steady-state endogenous protein states and data that establish direct molecular links.

The manuscript lacks evidence for direct phosphorylation of MAP1B by GSK3beta, and a direct role of MAP1B in the tyrosination/detyrosination, and furthermore, a direct role of MAP1B and the tyrosination cycle in interstitial branching. Even at the level of interpretation or a hypothetical model, it is unclear how these are linked. Typically, phosphorylation of MAPs results in their dissociation from microtubules - but the authors seem to suggest that MAP1B phosphorylation will enhance microtubule-binding, which in turn somehow would promote tyrosination, and the latter would again somehow promote branching. The manuscript would benefit from evidence in support of: 1) direct phosphorylation of MAP1B by GSK3beta, 2) MAP1B or MAP1-B interaction with TTL, and/or 3) enhanced microtubule dynamics using EB3 or another microtubule plus end marker at sites of interstitial branch formation - if the model is that more dynamic microtubules promote axon branching.

Major Comments/Concerns

1) The authors use a triple residue (S1260/T1265/S1391) phosphomimetic MAP1B mutant called "mode I phosphorylated MAP1B". The S1391 is primed by Dyrk1 phosphorylation of S1395. It is unclear whether these have been experimentally validated using direct phosphorylation assays (i.e., in vitro phosphorylation of recombinant MAP1B and corresponding mutants). Are these predicted sites or validated sites? Confidence in the claim of functional MAP1B regulation by GSK3beta is low without any evidence for direct phosphorylation.

In addition, the authors need to better explain (and show in a figure) in which domain these phosphorylation residues are located. Are they on a microtubule-binding domain? What kind of protein interactions are posited to affect?

2) The authors demonstrate using knock-in approach that GSK3beta is expressed in layer 2/3 neurons and their axons. However, is GSK3beta activated in these neurons in P0-P14? GSK3beta is considered to be constitutively active and its deactivation affects biological processes. Some evidence for GSK3beta activation would strengthen the claim/conclusion that "GSK3beta activation promotes interstitial branching through MAP1B phosphorylation". Can the authors stain with antibodies that recognize endogenous activated GSK3beta? Are there also any antibodies that detect phosphorylated MAP1B? To quantitatively show enhanced phosphorylation at the developmental window that interstitial branches form? This type of

evidence from a steady state physiological context and endogenous proteins is critical for supporting the conclusions.

3) The authors state that previous work has shown that phosphorylated MAP1B increases the ratio of tyrosinated to detyrosinated tubulin. This serves as rationale for experimentally testing the effects of tyrosination/detyrosination on interstitial branching. The link between MAP1B and tyrosination is confounding. Is tyrosinated tubulin a marker of more dynamic/less stable microtubules, which results from MAP1B dissociation from microtubules (MAPs typically "stabilize" microtubules). Or is MAP1B directly involved in the tyrosination of microtubules by scaffolding/interacting with tyrosine ligase? Evidence toward answering these two questions is critical in support of regulation of tyrosination via MAP1B.

As above, use of antibodies against endogenous tyrosinated/de-tyrosinated tubulin and quantifying levels in the respective developmental windows and in layer 2/3 neurons will provide much-needed steady-state physiological evidence for developmental enhancement of tubulin tyrosination during the formation of interstitial axon branches.

4) How does tubulin tyrosination promote interstitial branching? Or even MAP1B phosphorylation? If the underlying implication is that enhanced microtubule dynamics enable the formation of interstitial branches, this can be experimentally demonstrated by imaging microtubule dynamics at sites of branch formation in either slices or cultured dissociated neurons.

Referee #2:

In their manuscript "MAP1B Regulates Cortical Neuron Interstitial Axon Branching Through the Tubulin Tyrosination Cycle" Ziak et al. propose that collateral projecting neuron branching is regulated by a GSK3 β /MAP1B pathway that affects microtubuli stability by influencing MT-tyrosination. They find that overexpression of active GSK3 β leads to formation of ectopic branches in layer 4, in which wildtype CPNs do not form branches. Using state of the art in vivo techniques they show that GSK3 β phosphorylates MAP1B, which in turn leads to more tyrosinated, thus more stable, MTs. The work is conducted to a high scientific standard and text as well as images are well prepared. The GSK3 β /MAP1B pathway and its role in branch formation has been described before in vitro in dorsal root ganglia neurons, but this is -to my knowledge- the first evidence of the pathway and its effect on branching in vivo. Thus, following a few minor corrections laid out below, I endorse the manuscripts publication in the EMBO journal. I really would like to stress that this is an beautiful example of molecularly dissecting the a signaling pathway on axon branching in vivo.

- Line 112: In utero screen needs to be explained in more detail.
- Line 128: Tamoxifen-dependent bipartite system should be explained in the manuscript.
- Line 139: I do not think these plots make the data clearer, a scatter plot of animals pooled by condition might.
- Line 143: Is GSK3 β -CA here expressed at E15.5 or at P1, following tamoxifen treatment?
- Line 160: Figures are referred to incorrectly.
- Line 180: LOF is introduced as a variable and then never used again.
- Fig EV3B: I do not think this proof of concept is necessary.
- Fig EV3C/D: I would deem this impactful enough to be included in the main manuscript.
- Fig EV4A: Please show a ratio of tyrosinated tubulin to total tubulin, to verify that layer V is not just higher in total tubulin.
- Fig 4B the comparison of intensity between RFP/GFP and Dapi here does not seem very informative to me. Please show the intensity readings between GFP and Tyr-RFP. Same for the scatter plot. The ratio seems less informative than depicting the intensity readings against each other.
- Line 332: CRIPSR.

Referee #3:

The Ziak et al manuscript addresses the question of the molecular mechanisms at play in the regulation of interstitial axonal branching. Accurate axonal branching ensures suitable circuit formation and thus correct brain function. The present study tackles molecular control mechanisms underlying of one type of branching mechanism, interstitial branching, which is the "de novo" ramification of an established axon shaft. Interstitial branching occurs further away from the growth cone, and later in development, than the terminal branching, the most studied one. Using an elaborate experimental system, which allows to visualise and manipulate some of the layer 2/3 excitatory cortical neurons the authors show that interstitial branching can be initiated by GSK3 β -dependent phosphorylation of MAP1B. MAP1B, a microtubule-associated protein, shifts the balance of detyrosinated-tyrosinated tubulin towards tyrosinated tubulin, thereby promoting collateral branching.

The question addressed by this work is important and potentially of interest for a broad community of cell- and neuro-biologists, as perturbed axonal branching patterns can lead to abnormal circuit formation and brain malfunction. Furthermore, the experimental system used here allows to investigate this question in vivo, which is a huge strength of this approach, as the extracellular environment, absent in in-cellulo approaches with dissociated primary neuronal cultures, is expected to strongly

impact neuronal development and branching. The manuscript is written very clearly, and the figures are of very high quality.

Nevertheless, the manuscript contains certain weaknesses, which the authors must address before this work can be considered for publication.

Main points:

1. As the role of detyrosination/tyrosination in axon branching and the link of this tubulin modification to MAP1B present the most exciting, novel finding of this manuscript, it will be important to provide strong evidence for these results. However, it appears that the quantification of tyrosinated/detyrosinated tubulin levels in neurons (Fig 4B, EV4A), is not convincing.

a. The authors assessed changes in the levels of tyrosinated tubulin using the anti-tyrosinated-tubulin antibody in immunohistology, and the tyrosinated-tubulin sensor in single cells. It is very hard from these images to determine changes in tyrosination because a convincing counter-staining is missing. The authors could stain for the reciprocal tubulin modification - detyrosination (they might want to use the relatively novel monoclonal antibody (RM444), which appears to work better than the established rabbit polyclonal). Counterstaining tyr- and detyr-tubulin has a great advantage of contrasting potential changes, as the increase of one signal should be reflected in the decrease of the other. The authors could also test for delta2-tubulin, which is the follow-up modification of detyr-tubulin, and might increase if detyr-tubulin increases. If all of this fails, the authors need to at least provide a counterstaining with a general tubulin antibody to provide a ration total tubulin vs. tyr-tubulin.

b. Fig EV4A: What is the specific signal in this staining? There is a background signal and some axon-like structures whose staining seem more specific. While the background staining seems stronger in layer V (which could be because there is more "in between nuclei" space which is stained with the antibody), there seems to be more strongly labelled axon-like structures in layer IV.

c. As high detyr-tub and low tyr-tub repress branching, why decreasing tyrosinated tubulin (OE VASH) didn't reduce layer 5 branching?

2. The authors propose that MAP1B binding to detyrosinated microtubules and stabilising them restricts axon branching. MAP1B phosphorylation and detachment from microtubules shifts the equilibrium towards tyrosinated microtubules, destabilises microtubules allowing for the branch to be formed. To bolster the hypothesis that detyrosination of tubulin is a specific signal, the authors need to show whether other PTMs (K40 acetylation, polyglutamylation) are not affected following MAP1B phosphorylation.

Minor comments:

1. Tagging the full-length MAP1B C-terminally would result in visualizing only the light chain of this protein: how do the authors ensure that the MAP1B heavy chain does not behave different?

2. Authors should show the distribution of total and P-MAP1B in the neurons. One would expect to see a difference in the abundance of MAP1B-P between layers 4 and 5.

3. The discussion would benefit from being more concise: it constitutes a quarter of the length of the main manuscript. The authors should further address the following points:

a. While reference 70 indeed shows that katanin has a preference for tyrosinated microtubules, the authors also demonstrate that this preference is overwritten by polyglutamylation. In neurons, polyglutamylation being a major modification on microtubules, it is difficult to imagine it would not overshadow the effect of tyrosination.

b. Furthermore, the discussion of katanin data (not shown), should be rather moved to the main text of the paper, and not only mentioned in the discussion. In general, it is not good practise to discuss non-shown data, the authors should think about omitting this point, or show the data.

c. The authors should comment on possible different mechanisms involved in interstitial vs. terminal axon branching.

Dear Dr. Gailite,

Thank you for your interest in our manuscript and for inviting us to submit a revised version of our study. We appreciate your general summary, which has helped us to prioritize new experiments. In addition to the detailed point-by-point response to the reviewers below, we also provide comments directed toward the three key points you stressed in your summary. We believe that, taken together, these revisions strengthen our study and address the reviewers' central concerns.

Sincerely,

Jakub Ziak and Alex L. Kolodkin (on behalf of all authors)

A. GENERAL COMMENTS FROM THE EDITOR.

1 - Reviewers #1 and #3 find that direct evidence for MAP1B phosphorylation by GSK3beta would need to be provided.

We agree that this is an important point. Fortunately, direct evidence for MAP1B phosphorylation by GSK3 β has been shown previously by others (Barnat et al., 2016; Goold et al., 1999; Goold and Gordon-Weeks, 2005, 2001; Scales et al., 2009; Trivedi et al., 2005); we now state this in our revised manuscript and cite the relevant references. We also investigated MAP1B phosphorylation at Threonine 1265 (pT1265-MAP1B) and GSK3 β activation in cortical neurons *in vivo* (using the only available antibody that recognizes one of the MAP1B mode I phosphoresidues, and another antibody that recognizes non-phosphorylated GSK3 β , developed by Grabinski&Kanaan (Grabinski and Kanaan, 2016)). We performed immunolabeling of brain slices derived from postnatal day 4 (P4) mice (e.g. the age when interstitial axon branches are initiated in the layer 2/3 CPNs). Our results show that pT1265-MAP1B is enriched in axons in deeper cortical layers, and that GSK3 β is activated in cortical layer 5 neuropil.

These new data are presented in **Figure 1F and Figure EV3F**.

2 - The relationships between MAP1B, tubulin tyrosination and microtubule dynamics need further clarification.

We agree that this is also an important point. However, we would like to point out that the major scope of our paper is the identification of interstitial axon branching molecular regulators *in vivo*, which, to our knowledge, are almost completely unknown. To achieve this goal, we have developed a tractable system for sparse and robust gene targeting and neuronal labeling *in vivo* that allows us to investigate this problem. This present study is one of the first that systematically addresses cytosolic components of axon collateral formation in the mammalian cerebral cortex.

Though our central goal is not the provision of detailed biochemical evidence for the regulation of tubulin tyrosination by MAP1B, we agree that additional evidence would strengthen our study. Therefore, we have performed additional investigation of axonal microtubule dynamics using *in vitro* approaches and time-lapse microscopy. Our results show that velocity of EB3-GFP proteins (a measure of microtubule dynamics) decreases following acute (1 – 4 hours) pharmacological inhibition of GSK3 β , which is in line with our *in vivo* observations.

We have updated the corresponding sections of our Methods, and we added a new co-author (Xyniu Jin) who participated in this experiment. These new data are presented in **Figure 6, Appendix Figure S8 and Movie EV1**.

3 – The reviewers #2 and #3 point out that a better quantification of tubulin tyrosination would be required.

We agree that this is important, and so we have performed additional experiments using the tyrosination sensor, normalizing the signal to the total level of α -tubulin. We find that tyrosination is increased in layer 2/3 CPN axons and in layer 5, as compared to layer 4. In addition, we designed a construct for tagging endogenous α -tubulin in cortical neurons to investigate its axonal distribution. In brief, we tagged Tuba1 β with Hemagglutinin (HA) at its N-terminus (such that the C-terminus remain accessible for detyrosination), and we found that levels of HA-tubulin are equivalent in axons passing through layers 4 and 5. Together, these data support our initial results, and they are now presented in **Figure 4B and in Figure EV4**.

B. COMMENTS FROM REFEREES.

Referee #1:

Ziak et al investigate interstitial axon branching in layer 2/3 callosal projection neurons (CPNs) within the mammalian brain. The study employs a unique approach that enables precise labeling of these neurons, allowing for both quantitative analysis of axonal morphology and targeted gene expression manipulation. The study's central finding is that the serine/threonine kinase GSK3 β plays a pivotal role in promoting interstitial axon branching in layer 2/3 CPNs. Activation of GSK3 β triggers a cascade of events that leads to the release of MAP1B-mediated inhibition of axon branching. This molecular pathway is proposed to serve as a "brake" on axon branching, which can be released by GSK3 β activation. Furthermore, the study identifies that GSK3 β /MAP1B signaling influences the tyrosination/detyrosination cycle of α -tubulin, which is proposed to be part of the underlying mechanism of interstitial axon branch regulation.

Technically and analytically, the paper is strong with excellent imaging, great genetic tools, sound analysis and statistics. The data are clear. Conceptually, however, the links between GSK3beta, MAP1B, tyrosination/detyrosination and interstitial branching are tenuous, and the overall conclusions/models proposed are based on artificial manipulations with very little evidence with steady-state endogenous protein states and data that establish direct molecular links.

RESPONSE: We thank the referee for her/his positive comments (“*the paper is strong with excellent imaging, great genetic tools, sound analysis and statistics*”), constructive criticism, and helpful suggestions. A major weakness this reviewer sees in our study is “...*very little evidence with steady-state endogenous protein states and data that establish direct molecular links.*” We agree that our original manuscript did not address biochemical principles governing interactions between the individual components of the GSK3 β -MAP1B pathway we describe. We would like to point out that the major scope of our paper is the identification *in vivo* of interstitial axon branching molecular regulators, which, to our knowledge are almost completely unknown. Our present study is one of the first to systematically addresses cytosolic components critical for interstitial axon collateral formation in the mammalian cerebral cortex. Importantly, many of our *in vivo* manipulations are informed by previous molecular data generated *in vitro* and referenced in our paper. These studies (Bouquet et al., 2004; Goold et al., 1999; Utreras et al., 2008) have already demonstrated interactions between GSK3 β , MAP1B, and tubulin tyrosination, and so we did not repeat these assessments here. Rather, we sought to investigate their function *in vivo* in the context of cortical neuron development.

The manuscript lacks evidence for direct phosphorylation of MAP1B by GSK3beta, and a direct role of MAP1B in the tyrosination/detyrosination, and furthermore, a direct role of MAP1B and the tyrosination cycle in interstitial branching. Even at the level of interpretation or a hypothetical model, it is unclear how these are linked.

RESPONSE: We apologize for not being clear enough in our manuscript on this point. Indeed, it has been shown previously that GSK3 β directly phosphorylates MAP1B (Goold et al., 1999; Trivedi et al., 2005), and in our study we clarify this further. The role of MAP1B in the regulation of tyrosination cycle has been previously analyzed in cultured cells *in vitro* (Goold et al., 1999; Utreras et al., 2008). Although it is certainly possible that MAP1B directly binds to TTL or detyrosination enzymes (one study suggests that TTL binds MAP1B independent of phosphorylation, (Utreras et al., 2008)), an alternative hypothesis is that the effect of MAP1B is indirect through modulation of other microtubule posttranslational modifications (Barnat et al., 2016; Ebberink et al., 2023). Lastly, it is also possible that the regulation is indirect through the accessibility of the microtubule lattice by detyrosinating enzymes, and that phosphorylated MAP1B protects microtubules from SVBP/VASH binding. We think that testing all of these possible hypotheses is beyond the scope of our current study, which focuses on linking axonal morphologies (interstitial axon branching) with specific signaling proteins. In our revised discussion, we have added possible explanations for how MAP1B and tyrosination cycle might be linked. Importantly, we believe that our data (presented in the original submission, now in revised **Figure 2, Figure 5 and Figure 6**) address the direct role of MAP1B and the tyrosination cycle in interstitial axon branching. This is owing to the evidence we obtained from our gain-of-function, loss-of-function, and mutagenesis experiments *in vivo* on layer 2/3 cortical neuron interstitial axon branching.

Typically, phosphorylation of MAPs results in their dissociation from microtubules - but the authors seem to suggest that MAP1B phosphorylation will enhance microtubule-binding, which in turn somehow would promote tyrosination, and the latter would again somehow promote branching.

RESPONSE: We agree that for many MAPs, phosphorylation is thought to be inactivating, or leads to dissociation of MAPs from microtubules. However, this has already been shown not to be the case for MAP1B (Sutherland, 2011; Zhou and Snider, 2005). Indeed, it has been proposed that MAP1B is activated by phosphorylation (Gobrecht et al., 2014; Scales et al., 2009; Trivedi et al., 2005), and that phosphorylation does not affect MAP1B binding to MTs (Goold et al., 1999; Sato-Yoshitake et al., 1989). To what extent phosphorylation of MAP1B influences its' binding to actin is not clear (Cueille et al., 2007; Pedrotti and Islam, 1996).

The manuscript would benefit from evidence in support of:

1) Direct phosphorylation of MAP1B by GSK3beta

RESPONSE: As described above, it has been already shown that GSK3 β phosphorylates MAP1B, and we cite these references in the manuscript. In addition, we now provide new data that assesses the distribution of phospho-MAP1B (pT1265-MAP1B) and total MAP1B in cortical slices using immunohistochemistry, showing that pT1265-MAP1B is enriched in axon bundles in cortical layer 5 (see below). These new data are presented in the **Figure EV3F**.

2) MAP1B or MAP1-B interaction with TTL

RESPONSE: As described above, there are multiple possibilities for how MAP1B could regulate tubulin tyrosination cycle. MAP1B and TTL interactions have been described in N2A and Cos7 cells *in vitro* (Utreras et al., 2008). In our revised Discussion, we now discuss more fully how MAP1B and the tyrosination cycle might be linked.

3) Enhanced microtubule dynamics using EB3 or another microtubule plus end marker at sites of interstitial branch formation - if the model is that more dynamic microtubules promote axon branching.

RESPONSE: We appreciate this comment and suggestion, and we have performed this experiment, as requested by this reviewer, in dissociated layer 2/3 CPNs (see below). In brief, our results show that acute pharmacological inhibition of GSK3 β decreases EB3-GFP velocity at axonal branch points. These new data are presented in the **Figure 6, Appendix Figure S8 and Movie EV1**.

Major Comments/Concerns

1) The authors use a triple residue (S1260/T1265/S1391) phosphomimetic MAP1B mutant called "mode I phosphorylated MAP1B". The S1391 is primed by Dyrk1 phosphorylation of S1395. It is unclear whether these have been experimentally validated using direct phosphorylation assays (i.e., in vitro phosphorylation of recombinant MAP1B and corresponding mutants). Are these predicted sites or validated sites? Confidence in the claim of functional MAP1B regulation by GSK3beta is low without any evidence for direct phosphorylation. In addition, the authors need to better explain (and show in a figure) in which domain these phosphorylation residues are located. Are they on a microtubule-binding domain? What kind of protein interactions are posited to affect?

RESPONSE: As we stated above, we apologize for not being clear on this important point in our original manuscript. Indeed, these phosphorylation-sites have been experimentally validated prior to our study. GSK3 β directly phosphorylates MAP1B at these specific residues (Goold and Gordon-Weeks, 2005; Scales et al., 2009; Trivedi et al., 2005), and we now clarify this in our revised manuscript. Further, we have included a new figure panel (**Fig. EV2E**) with the MAP1B domains, as suggested by the reviewer, and we reference studies that analyze the impact of mode I phosphorylation on the function and possible binding partners of MAP1B.

2) The authors demonstrate using knock-in approach that GSK3beta is expressed in layer 2/3 neurons and their axons. However, is GSK3beta activated in these neurons in P0-P14? GSK3beta is considered to be constitutively active and its deactivation affects biological processes. Some evidence for GSK3beta activation would strengthen the claim/conclusion that "GSK3beta activation promotes interstitial branching through MAP1B phosphorylation". Can the authors stain with antibodies that recognize endogenous activated GSK3beta? Are there also any antibodies that detect phosphorylated MAP1B? To quantitatively show

enhanced phosphorylation at the developmental window that interstitial branches form? This type of evidence from a steady state physiological context and endogenous proteins is critical for supporting the conclusions.

RESPONSE: We agree with the reviewer that this is an important point. To provide evidence for GSK3 β activation in axonal segments in layer 2/3 CPNs, we acquired a commercially available antibody directed against phospho-GSK3 β (pS9-GSK3 β , which is an inactive form of the enzyme) and an antibody directed against nonphospho-GSK3 β (np-GSK3, an active form of the enzyme) (described in: Grabinski and Kanaan, 2016), and we analyzed immunoreactivity using these antibodies in cortical slices derived from P4 cortex. Our data show that the ratio of np-GSK3 β /pS9-GSK3 β is increased in layer 5 as compared to layer 4 (Figure 1F).

Assessing the distribution of activated (phosphorylated) MAP1B is challenging since there is, to our knowledge, only one available phospho-specific MAP1B antibody (pT1265-MAP1B, (Trivedi et al., 2005)). However, we acquired this antibody and assessed the distribution of phospho-MAP1B along with total MAP1B antibody (as a control) in developing cortices at P4. Our results show that pT1265-MAP1B is enriched in neuropil in deeper cortical layers (5 and 6), as compared to layer 4. These new data are presented in the Figure EV3F. Collectively, our new observations are in line with our proposed model, suggesting that GSK3 β activity is locally increased in cortical layer 5 during the developmental period when interstitial axon branching is initiated.

3) The authors state that previous work has shown that phosphorylated MAP1B increases the ratio of tyrosinated to detyrosinated tubulin. This serves as rationale for experimentally testing the effects of tyrosination/detyrosination on interstitial branching. The link between MAP1B and tyrosination is confounding. Is tyrosinated tubulin a marker of more dynamic/less stable microtubules, which results from MAP1B dissociation from microtubules (MAPs typically "stabilize" microtubules). Or is MAP1B directly involved in the tyrosination of microtubules by scaffolding/interacting with tyrosine ligase? Evidence toward answering these two questions is critical in support of regulation of tyrosination via MAP1B. As above, use of antibodies against endogenous tyrosinated/de-tyrosinated tubulin and quantifying levels in the respective developmental windows and in layer 2/3 neurons will provide much-needed steady-state physiological evidence for developmental enhancement of tubulin tyrosination during the formation of interstitial axon branches.

RESPONSE: We appreciate that the rationale for testing the effect of tubulin tyrosination cycle on interstitial axon branching as a possible downstream effector of GSK3 β /MAP1B axis was clear to the reviewer in our original manuscript. Indeed, as mentioned above, our idea for this overall research direction was encouraged by previously published MAP1B biochemical data and tubulin tyrosination (Barnat et al., 2016; Goold et al., 1999; Tint et al., 2005), and thus we do not think it is necessary to repeat similar experiments as part of this present study. The reviewer specifically asks if "tyrosinated tubulin is a marker of more dynamic/less stable microtubules, which results from MAP1B dissociation from microtubules?" Although tyrosinated tubulin is considered to be a marker of more dynamic microtubules, this appears not to result from MAP1B dissociation from microtubules, as it has been demonstrated previously (Goold et al., 1999). The reviewer also asks if "MAP1B is directly involved in the tyrosination of microtubules by scaffolding/interacting with tyrosine ligase?" Indeed, the direct binding of MAP1B and TTL has been demonstrated (Utreras et al., 2008), yet there are multiple hypotheses, mentioned above, for how MAP1B could influence the tyrosination cycle. Finally, the reviewer suggests the "use of antibodies against endogenous tyrosinated/de-tyrosinated tubulin and quantifying levels in the respective developmental windows and in layer 2/3 neurons". As suggested, we performed immunostainings (using an antibody that recognizes detyrosinated tubulin as recommended by reviewer #3) to show the distribution of these tubulin posttranslational modifications (PTMs) in the cortex. We found that the ratio of tyrosinated/detyrosinated tubulins in the cortex is indeed increased in layer 5 in comparison to layer 4 (Appendix Figure S6A).

4) How does tubulin tyrosination promote interstitial branching? Or even MAP1B phosphorylation? If the underlying implication is that enhanced microtubule dynamics enable the formation of interstitial branches, this can be experimentally demonstrated by imaging microtubule dynamics at sites of branch formation in either slices or cultured dissociated neurons.

RESPONSE: We agree with this point and thank the reviewer for this suggestion. We assessed microtubule dynamics in developing layer 2/3 CPN axons *in vitro* at DIV4. E15.5 embryos were *in utero*

electroporated with plasmids encoding pCAG-mCherry and pCAG-EB3-GFP. At E18.5, neurons were dissociated from electroporated cortices and plated on glass-bottom cell culture plates. This approach decreases phenotypic variability of transfected neurons since the majority of electroporated neurons represent layer 2/3 CPNs. At DIV4, EB3-GFP was tracked under acute GSK3 β pharmacological inhibition (using the CHIR99021 inhibitor, (Bennett et al., 2002)). We found that GSK3 β inhibition decreases the speed of EB3-GFP at developing branch points and leads to a modest decrease of EB3 particle number at the sites of axon branches. These results show that inhibition of GSK3 β decreases microtubule dynamics, which is in line with our proposed model. Our new data are now presented in the Figure 6, Appendix Figure S8 and Movie EV1.

Referee #2:

In their manuscript "MAP1B Regulates Cortical Neuron Interstitial Axon Branching Through the Tubulin Tyrosination Cycle" Ziak et al. propose that collateral projecting neuron branching is regulated by a GSK3 β /MAP1B pathway that affects microtubuli stability by influencing MT-tyrosination. They find that overexpression of active GSK3 β leads to formation of ectopic branches in layer 4, in which wildtype CPNs do not form branches. Using state of the art in vivo techniques they show that GSK3 β phosphorylates MAP1B, which in turn leads to more tyrosinated, thus more stable, MTs. The work is conducted to a high scientific standard and text as well as images are well prepared. The GSK3 β /MAP1B pathway and its role in branch formation has been described before in vitro in dorsal root ganglia neurons, but this is -to my knowledge- the first evidence of the pathway and its effect on branching in vivo. Thus, following a few minor corrections laid out below, I endorse the manuscripts publication in the EMBO journal. I really would like to stress that this is an beautiful example of molecularly dissecting the a signaling pathway on axon branching in vivo.

RESPONSE: We thank the reviewer for their positive evaluation, appreciation of our study, and their helpful comments.

- Line 112: *In utero* screen needs to be explained in more detail.

RESPONSE: We have added text to explain our *in utero* screen further, as suggested. The paper describing this screen was recently published in J. Neurosci. (Dorskind et al., 2023), and we have updated this reference accordingly.

- Line 128: *Tamoxifen-dependent bipartite system should be explained in the manuscript.*

RESPONSE: We thank the reviewer for this suggestion. We have added a more detailed explanation to the 'Results' section. As mentioned above, the paper describing our development of this method recently published in J Neurosci., and we updated the reference accordingly.

- Line 139: *I do not think these plots make the data clearer, a scatter plot of animals pooled by condition might.*

RESPONSE: We thank the reviewer for this suggestion. However, we would prefer to keep these data presented as they are because we believe they better represent the overall impact of various experimental conditions on interstitial axon branching. Please note that scatter plots are shown for all experimental conditions in our supplemental information (Appendix Figure S1 – S5) to facilitate data transparency.

- Line 143: *Is GSK3 β -CA here expressed at E15.5 or at P1, following tamoxifen treatment?*

RESPONSE: The GSK3 β -CA plasmid is electroporated at E15.5, and the enzyme is expressed from P1 on; this is now clarified in our revised manuscript.

- Line 160: *Figures are referred to incorrectly.*

RESPONSE: We have corrected our mistake in this revised manuscript.

- Line 180: *LOF is introduced as a variable and then never used again.*

RESPONSE: Thank you for spotting this; LOF does seem redundant here.

- Fig EV3B: *I do not think this proof of concept is necessary.*

RESPONSE: The TKIT method is a relatively new experimental system (Fang et al., 2021) which was mainly optimized for AAV gene delivery. Therefore, we wanted to show that it can be used in combination with *in utero* electroporation as well.

- Fig EV3C/D: I would deem this impactful enough to be included in the main manuscript.

RESPONSE: We thank the reviewer for this suggestion; however, these data confirm what has been shown in previous studies (Uemura et al., 2016), and after revising our manuscript we find that we do not have enough space within our main figures. Therefore, we would prefer to leave these data in the EV figure (which will be available online within the main text and figures).

- Fig EV4A: Please show a ratio of tyrosinated tubulin to total tubulin, to verify that layer V is not just higher in total tubulin.

RESPONSE: We now present immunostaining for total α -tubulin in **Appendix Figure S6**. In addition, in the same figure we also assay distribution of de-tyrosinated tubulin to complement these data.

- Fig 4B the comparison of intensity between RFP/GFP and Dapi here does not seem very informative to me. Please show the intensity readings between GFP and Tyr-RFP. Same for the scatter plot. The ratio seems less informative than depicting the intensity readings against each other.

RESPONSE: We apologize for not being clear in our initial presentation. We had plotted the RFP/GFP ratio in the graph in the original manuscript (in our example, this was represented by the yellow line in our original Fig. 4B). In general, we used DAPI as a measure of cortical layers (decreased DAPI intensity indicates transition from layer 4 to layer 5) but DAPI is never used for data normalization. In our revised manuscript, we now normalize Tyr-T sensor data to the level of total tubulin, as suggested by the other reviewers. Our new measurements are in line with data from our original manuscript, and they are shown in **Figure 4B**.

- Line 332: CRIPSR.

RESPONSE: Thank you for spotting this error; we have corrected it.

Referee #3:

The Ziak et al manuscript addresses the question of the molecular mechanisms at play in the regulation of interstitial axonal branching. Accurate axonal branching ensures suitable circuit formation and thus correct brain function. The present study tackles molecular control mechanisms underlying of one type of branching mechanism, interstitial branching, which is the "de novo" ramification of an established axon shaft. Interstitial branching occurs further away from the growth cone, and later in development, than the terminal branching, the most studied one. Using an elaborate experimental system, which allows to visualise and manipulate some of the layer 2/3 excitatory cortical neurons the authors show that interstitial branching can be initiated by GSK3 β -dependent phosphorylation of MAP1B. MAP1B, a microtubule-associated protein, shifts the balance of de-tyrosinated-tyrosinated tubulin towards tyrosinated tubulin, thereby promoting collateral branching. The question addressed by this work is important and potentially of interest for a broad community of cell- and neuro-biologists, as perturbed axonal branching patterns can lead to abnormal circuit formation and brain malfunction. Furthermore, the experimental system used here allows to investigate this question in vivo, which is a huge strength of this approach, as the extracellular environment, absent in in-cellulo approaches with dissociated primary neuronal cultures, is expected to strongly impact neuronal development and branching. The manuscript is written very clearly, and the figures are of very high quality. Nevertheless, the manuscript contains certain weaknesses, which the authors must address before this work can be considered for publication.

RESPONSE: We thank the reviewer for their appreciation of the importance of understanding how cortical interstitial axon branching is regulated and for their helpful comments and suggestions.

Main points:

1. As the role of de-tyrosination/tyrosination in axon branching and the link of this tubulin modification to MAP1B present the most exciting, novel finding of this manuscript, it will be important to provide strong evidence for these results. However, it appears that the quantification of tyrosinated/de-tyrosinated tubulin levels in neurons (Fig 4B, EV4A), is not convincing.

RESPONSE: We thank the reviewer for this comment. We agree that the function of the GSK3 β -MAP1B axis and tubulin tyrosination cycle in interstitial axon branching *in vivo* represents the major finding of

the study. The rationale for investigating tubulin tyrosination cycle was based on previous studies that have already shown the link between MAP1B and tyrosination. As suggested by the reviewer, we focus in our revision on strengthening the quantification of these levels, and also other tubulin PTMs levels, in neurons. **Figure 4B, Figure EV4** and **Appendix Figure S6 and S7** are now updated with our new data.

a. The authors assessed changes in the levels of tyrosinated tubulin using the anti-tyrosinated-tubulin antibody in immuno-histology, and the tyrosinated-tubulin sensor in single cells. It is very hard from these images to determine changes in tyrosination because a convincing counter-staining is missing. The authors could stain for the reciprocal tubulin modification - detyrosination (they might want to use the relatively novel monoclonal antibody (RM444), which appears to work better than the established rabbit polyclonal). Counterstaining tyr- and detyr-tubulin has a great advantage of contrasting potential changes, as the increase of one signal should be reflected in the decrease of the other. The authors could also test for delta2-tubulin, which is the follow-up modification of detyr-tubulin, and might increase if detyr-tubulin increases. If all of this fails, the authors need to at least provide a counterstaining with a general tubulin antibody to provide a ration total tubulin vs. tyr-tubulin.

RESPONSE: This is an extremely valuable point, and we welcome the suggestion to use this antibody. Indeed, the reason why we did not include detyrosination staining in the original manuscript was the lack of a working and reliable antibody. In our revised manuscript we now assess the tyrosination/detyrosination ratio in the cortex and find that it is increased in layer 5 in comparison to layer 4, which is in agreement with our original findings. These data are now shown in **Appendix Figure S6A**. In addition, we assessed the distribution of total α -tubulin using immunostaining (**Appendix Figure S6B and S6C**) and TKIT tagging (**Figure EV4A**). Finally, we provide more accurate quantification of tyrosination levels in individual neurons using the fluorescent tyrosination sensor (data are now normalized to the level of α -tubulin, **Figure 4B**), and additional validation of the sensor (**Figure EV4B**).

b. Fig EV4A: What is the specific signal in this staining? There is a background signal and some axon-like structures whose staining seem more specific. While the background staining seems stronger in layer V (which could be because there is more "in between nuclei" space which is stained with the antibody), there seems to be more strongly labelled axon-like structures in layer IV.

RESPONSE: We think that the axon-like structures that reviewer is refereeing to are apical dendrites of layer 5 pyramidal neurons extending through the layers 5 and 4. The 'background' signal is in fact what we quantified in this staining, and it likely represents the cumulative staining in axons and basal dendrites. Indeed, this signal is stronger in layer 5, as the reviewer notes. We now support these conclusions in this revision by co-staining these slices with antibodies directed against tau (axonal marker), shown in the **Appendix Figure S6D**. In addition, we now also present MAP1B staining in the cortex (**Figure EV3F**), which strongly labels cell bodies and apical dendrites, revealing similar structures as the reviewer noticed in the original Figure EV4A. In conclusion, we moved these data from our original Figure EV4A into a new **Appendix Figure S6**.

c. As high detyr-tub and low tyr-tub repress branching, why decreasing tyrosinated tubulin (OE VASH) didn't reduce layer 5 branching?

RESPONSE: The reviewer is correct – we did not observe decreased interstitial axon branching in experimental conditions where detyrosination was promoted (VASH GOF and TTL LOF). We only observed suppression of interstitial axon branching with TTL LOF when neurons were primed to branch with GSK3 β -CA overexpression (**Fig. 6C** in revised manuscript). Therefore, it seems that the levels of remaining tyrosinated tubulin in VASH OE and TTL LOF conditions are enough to support interstitial axon branching in the layer 5. By 'remaining tyrosinated tubulins' we mean the newly synthesized proteins that naturally contain a tyrosine residue at their C termini and can only be detyrosinated after incorporation into the microtubule lattice. Alternatively, it is also possible that another tubulin PTM is epistatic to tyrosination/detyrosination (as suggested by the reviewer below).

2. The authors propose that MAP1B binding to detyrosinated microtubules and stabilising them restricts axon branching. MAP1B phosphorylation and detachment from microtubules shifts the equilibrium towards tyrosinated microtubules, destabilises microtubules allowing for the branch to be formed. To bolster the hypothesis that detyrosination of tubulin is a specific signal, the authors need to show whether other PTMs (K40 acetylation, polyglutamylaton) are not affected following MAP1B phosphorylation.

RESPONSE: We thank the reviewer for raising this important point. Indeed, it is possible that the axon branch-promoting signals downstream of GSK3 β are translated by other molecules than MAP1B, or that other

tubulin PTMs are affected following MAP1B phosphorylation. Previous studies have already investigated the later point and found that phosphorylated MAP1B promotes tyrosination over detyrosination, as described above in response to reviewers #1 and #2. MAP1B also promotes tubulin acetylation (Barnat et al., 2016; Bouquet et al., 2004). We used polyE antibody to detect polyglutamylation in cortical neurons overexpressing GSK3 β -CA *in vivo* in layer 2/3 CPNs, and both polyE and Ac-Tubulin (to detect K40 acetylation) in cortical neurons lacking MAP1B. Our results suggest that K40 acetylation and polyglutamylation are not affected by GSK3 β -CA overexpression or MAP1B deletion (Appendix Figure S7). In addition, we have also assessed the spatial distribution of α -tubulin and other tubulin PTMs (polyglutamylation, acetylation) in developing axons *in vivo* at P4 cortical slices. Collectively, we find that polyglutamylated and acetylated tubulin distribution seems not to reflect cortical layering, as we do not observe layer-specific distribution of these two PTMs in the cortex at P4. These PTMs are also notably enriched in axon bundles. On the other hand, tyrosination is the PTM with the highest layer 5/layer 4 ratio, as described above. These new data are now presented in the Appendix Figure S6.

Minor comments:

1. *Tagging the full-length MAP1B C-terminally would result in visualizing only the light chain of this protein: how do the authors ensure that the MAP1B heavy chain does not behave differently?*

RESPONSE: We thank the reviewer for raising this point. The rationale for this tagging experiment was to ask whether MAP1B is evenly distributed in axons or specifically enriched at axonal branch points. We now show (Figure EV3F) the distribution of MAP1B using a MAP1B mouse monoclonal antibody AA6, which recognizes the MAP1B heavy chain. The staining is strongest in the neuronal body and apical dendrite, with the faint signal detectable in axons.

2. *Authors should show the distribution of total and P-MAP1B in the neurons. One would expect to see a difference in the abundance of MAP1B-P between layers 4 and 5.*

RESPONSE: We now show the distribution of both MAP1B and p-MAP1B, as suggested, and these new data are shown in the Figure EV3F. Indeed, we observe an increase in the abundance of pMAP1B in deeper cortical layers.

3. *The discussion would benefit from being more concise: it constitutes a quarter of the length of the main manuscript. The authors should further address the following points:*

a. *While reference 70 indeed shows that katanin has a preference for tyrosinated microtubules, the authors also demonstrate that this preference is overwritten by polyglutamylation. In neurons, polyglutamylation being a major modification on microtubules, it is difficult to imagine it would not overshadow the effect of tyrosination.*

RESPONSE: We thank the reviewer for this interesting comment – this could also explain the rather small changes we see with tyrosination cycle manipulation. We now discuss possible involvement of other tubulin PTMs on the regulation of axonal morphology in cortical neurons. Indeed, our immunohistochemistry results indicate PolyE signal is mainly found in the axons in cortical wall. In our future experiments, we will address the function of tubulin polyglutamylation on interstitial axon branching in a greater detail.

b. *Furthermore, the discussion of katanin data (not shown), should be rather moved to the main text of the paper, and not only mentioned in the discussion. In general, it is not good practise to discuss non-shown data, the authors should think about omitting this point, or show the data.*

RESPONSE: We have now added an additional supplemental figure (Appendix Figure S9) to show these data, as suggested.

c. *The authors should comment on possible different mechanisms involved in interstitial vs. terminal axon branching.*

RESPONSE: We discuss these possible differences in our revised discussion.

REFERENCES

- Barnat, M., Benassy, M.-N., Vincensini, L., Soares, S., Fassier, C., Propst, F., Andrieux, A., von Boxberg, Y., Nothias, F., 2016. The GSK3-MAP1B pathway controls neurite branching and microtubule dynamics. *Mol. Cell. Neurosci.* 72, 9–21. <https://doi.org/10.1016/j.mcn.2016.01.001>
- Bennett, C.N., Ross, S.E., Longo, K.A., Bajnok, L., Hemati, N., Johnson, K.W., Harrison, S.D., MacDougald, O.A., 2002. Regulation of Wnt signaling during adipogenesis. *J. Biol. Chem.* 277, 30998–31004. <https://doi.org/10.1074/jbc.M204527200>
- Bouquet, C., Soares, S., von Boxberg, Y., Ravaille-Veron, M., Propst, F., Nothias, F., 2004. Microtubule-associated protein 1B controls directionality of growth cone migration and axonal branching in regeneration of adult dorsal root ganglia neurons. *J. Neurosci. Off. J. Soc. Neurosci.* 24, 7204–7213. <https://doi.org/10.1523/JNEUROSCI.2254-04.2004>
- Cueille, N., Blanc, C.T., Popa-Nita, S., Kasas, S., Catsicas, S., Dietler, G., Riederer, B.M., 2007. Characterization of MAP1B heavy chain interaction with actin. *Brain Res. Bull.* 71, 610–618. <https://doi.org/10.1016/j.brainresbull.2006.12.003>
- Dorskind, J.M., Sudarsanam, S., Hand, R.A., Ziak, J., Amoah-Dankwah, M., Guzman-Clavel, L., Soto-Vargas, J.L., Kolodkin, A.L., 2023. Drebrin Regulates Collateral Axon Branching in Cortical Layer II/III Somatosensory Neurons. *J. Neurosci.* 43, 7745–7765. <https://doi.org/10.1523/JNEUROSCI.0553-23.2023>
- Ebberink, E., Fernandes, S., Hatzopoulos, G., Agashe, N., Chang, P.-H., Guidotti, N., Reichart, T.M., Reymond, L., Velluz, M.-C., Schneider, F., Pourroy, C., Janke, C., Gönczy, P., Fierz, B., Aumeier, C., 2023. Tubulin engineering by semi-synthesis reveals that polyglutamylation directs detyrosination. *Nat. Chem.* 15, 1179–1187. <https://doi.org/10.1038/s41557-023-01228-8>
- Fang, H., Bygrave, A.M., Roth, R.H., Johnson, R.C., Haganir, R.L., 2021. An optimized CRISPR/Cas9 approach for precise genome editing in neurons. *eLife* 10, e65202. <https://doi.org/10.7554/eLife.65202>
- Gobrecht, P., Leibinger, M., Andreadaki, A., Fischer, D., 2014. Sustained GSK3 activity markedly facilitates nerve regeneration. *Nat. Commun.* 5, 4561. <https://doi.org/10.1038/ncomms5561>
- Goold, R.G., Gordon-Weeks, P.R., 2005. The MAP kinase pathway is upstream of the activation of GSK3beta that enables it to phosphorylate MAP1B and contributes to the stimulation of axon growth. *Mol. Cell. Neurosci.* 28, 524–534. <https://doi.org/10.1016/j.mcn.2004.11.005>
- Goold, R.G., Gordon-Weeks, P.R., 2001. Microtubule-associated protein 1B phosphorylation by glycogen synthase kinase 3beta is induced during PC12 cell differentiation. *J. Cell Sci.* 114, 4273–4284. <https://doi.org/10.1242/jcs.114.23.4273>
- Goold, R.G., Owen, R., Gordon-Weeks, P.R., 1999. Glycogen synthase kinase 3beta phosphorylation of microtubule-associated protein 1B regulates the stability of microtubules in growth cones. *J. Cell Sci.* 112 (Pt 19), 3373–3384. <https://doi.org/10.1242/jcs.112.19.3373>
- Grabinski, T., Kanaan, N.M., 2016. Novel Non-phosphorylated Serine 9/21 GSK3β/α Antibodies: Expanding the Tools for Studying GSK3 Regulation. *Front. Mol. Neurosci.* 9, 123. <https://doi.org/10.3389/fnmol.2016.00123>
- Pedrotti, B., Islam, K., 1996. Dephosphorylated but not phosphorylated microtubule associated protein MAP1B binds to microfilaments. *FEBS Lett.* 388, 131–133. [https://doi.org/10.1016/0014-5793\(96\)00520-0](https://doi.org/10.1016/0014-5793(96)00520-0)
- Sato-Yoshitake, R., Shiomura, Y., Miyasaka, H., Hirokawa, N., 1989. Microtubule-associated protein 1B: molecular structure, localization, and phosphorylation-dependent expression in developing neurons. *Neuron* 3, 229–238. [https://doi.org/10.1016/0896-6273\(89\)90036-6](https://doi.org/10.1016/0896-6273(89)90036-6)
- Scales, T.M.E., Lin, S., Kraus, M., Goold, R.G., Gordon-Weeks, P.R., 2009. Nonprimed and DYRK1A-primed GSK3 beta-phosphorylation sites on MAP1B regulate microtubule dynamics in growing axons. *J. Cell Sci.* 122, 2424–2435. <https://doi.org/10.1242/jcs.040162>
- Sutherland, C., 2011. What Are the bona fide GSK3 Substrates? *Int. J. Alzheimers Dis.* 2011, 505607. <https://doi.org/10.4061/2011/505607>
- Tint, I., Fischer, I., Black, M., 2005. Acute inactivation of MAP1b in growing sympathetic neurons destabilizes axonal microtubules. *Cell Motil. Cytoskeleton* 60, 48–65. <https://doi.org/10.1002/cm.20045>
- Trivedi, N., Marsh, P., Goold, R.G., Wood-Kaczmar, A., Gordon-Weeks, P.R., 2005. Glycogen synthase kinase-3beta phosphorylation of MAP1B at Ser1260 and Thr1265 is spatially restricted to growing axons. *J. Cell Sci.* 118, 993–1005. <https://doi.org/10.1242/jcs.01697>
- Uemura, T., Mori, T., Kurihara, T., Kawase, S., Koike, R., Satoga, M., Cao, X., Li, X., Yanagawa, T., Sakurai, T., Shindo, T., Tabuchi, K., 2016. Fluorescent protein tagging of endogenous protein in brain neurons using

CRISPR/Cas9-mediated knock-in and in utero electroporation techniques. *Sci. Rep.* 6, 35861.
<https://doi.org/10.1038/srep35861>

Utreras, E., Jiménez-Mateos, E.M., Contreras-Vallejos, E., Tortosa, E., Pérez, M., Rojas, S., Saragoni, L., Maccioni, R.B., Avila, J., González-Billault, C., 2008. Microtubule-associated protein 1B interaction with tubulin tyrosine ligase contributes to the control of microtubule tyrosination. *Dev. Neurosci.* 30, 200–210. <https://doi.org/10.1159/000109863>

Zhou, F.-Q., Snider, W.D., 2005. GSK-3 β and Microtubule Assembly in Axons. *Science* 308, 211–214.
<https://doi.org/10.1126/science.1110301>

Dear Alex,

Thank you for submitting a revised version of your manuscript. Your study has now been seen by all original referees, who find that their previous concerns have been addressed and now recommend acceptance of the manuscript.

There now remain only a few editorial points, in addition to the minor comments by reviewer #3, that need addressing before I can extend acceptance of the manuscript:

1. CRediT has replaced the traditional author contributions section because it offers a systematic, machine-readable author contributions format that allows for more effective research assessment. Please remove the Authors Contributions from the manuscript and use the free text boxes beneath each contributing author's name in our online submission system to add specific details on the author's contribution. More information is available in our guide to authors.
2. Please rename "Conflict of interest" section into "Disclosure and competing interests statement" (further info: <https://www.embopress.org/page/journal/14602075/authorguide#conflictsofinterest>).
3. There are references to "data not shown" in lines 306, 332, 401, 467. According to our policy, which does not permit references to "data not shown", please include this information in the Appendix. Please see also <https://www.embopress.org/page/journal/14602075/authorguide#unpublisheddata>.
4. Please update the movie nomenclature to Movie EV1 and update the callouts accordingly. The legend should be removed from the manuscript text file and zipped with the movie file. Further information is available here: <https://www.embopress.org/page/journal/14602075/authorguide#expandedview>
5. Figure panels 4D-E are not mentioned in the manuscript text. There is a callout for figure panel 7B, which does not exist.
6. In the legend of the Appendix Figure S8, please update the reference to the reuse of EB3-GFP kymographs to Figure 6 (currently Figure 7).
7. Our data editors have flagged the following issues in figure legends that need correcting:
 - Please note that information related to n is missing in the legends of figures EV 1h; EV 4b; EV 5a-b.
 - Please define the error bars in the legends of figure 4b; EV 1h; EV 4b.
 - Please define the white arrowhead in the legend of figure 6a; EV 3c-d;.EV 4a.
8. Papers published in The EMBO Journal are accompanied online by a 'Synopsis' to enhance discoverability of the manuscript. Please submit a short (1-2 sentences) summary of the findings and their significance in addition to the already provided bullet points highlighting the key results. Please also send us a synopsis image that is 550x300-600 pixels large (width x height, jpeg or png format). You can either show a model or key data in the synopsis image. Please note that the image size is rather small and that text needs to be readable at the final size.

With best wishes,

leva

leva Gailite, PhD
Senior Scientific Editor
The EMBO Journal
Meyerhofstrasse 1
D-69117 Heidelberg
Tel: +4962218891309
i.gailite@embojournal.org

We realize that it is difficult to revise to a specific deadline. In the interest of protecting the conceptual advance provided by the work, we recommend a revision within 3 months (22nd Apr 2024). Please discuss the revision progress ahead of this time with the editor if you require more time to complete the revisions.

Referee #1:

The authors have addressed my comments and concerns. I have no further comments.

Referee #2:

The authors have addressed of my comments adequately. It is ready for publication. Congrats to a very nice paper.

Referee #3:

I would like to thank the authors for taking into consideration most of my (and other reviewer's) suggestions and adapting the manuscript accordingly. As I said earlier, this is a beautiful work using challenging techniques, addressing an important question in an in vivo setting. With the few minor changes (see below), I support the publication of this manuscript in the EMBO Journal.

1. Quantification of the changes of different tubulin PTMs in layer 4 and 5 (Appendix Fig. 6C): a better readout would be to plot the ratio PTM/total tubulin for layer 4 and 5, or the ratio (layer5/layer4) the ratios PTM/total tubulin (analogous to the graph in the Appendix Fig. S6A).
2. The authors should comment on the fact that while tyrosinated tubulin levels increase in layer 5, there is no concomitant decrease (relative to the total tubulin - see point 1 above) of the "opposite" tubulin form, the detyrosinated one (around the line 270).
3. Line 28: "interstitial" sounds very specific so early in the abstract: could you keep "axon branching" here, and only introduce "intersitial" in the 3rd sentence?
4. Line 296: "counteracting the normally occurring detyrosination of tubulin in neurons" would be more precise than "favouring increased tyrosinated MTs".
5. The discussion is still very long and could be made more precise. Please revise sentences which are long and confusing:
Line 399: the sentence "However, ...",
Lines 423-430: this sounds like a repetition of the result section and thus could be shortened.

The authors addressed the minor editorial issues.

Dear Alex,

Thank you for addressing the final editorial points. I am now pleased to inform you that your manuscript has been accepted for publication.

I will look into the synopsis text in the next couple of days and let you know if any edits to the journal style are needed.

If you have any questions, please do not hesitate to contact the Editorial Office. Thank you for this contribution to The EMBO Journal and congratulations on a successful publication!

With best wishes,

leva

leva Gailite, PhD
Senior Scientific Editor
The EMBO Journal
Meyerhofstrasse 1
D-69117 Heidelberg
Tel: +4962218891309
i.gailite@embojournal.org
